# Epigenetically reprogrammed methylation landscape drives the DNA self-assembly and serves as a universal cancer biomarker

Abu Ali Ibn Sina [1], Laura G. Carrascosa[1], Ziyu Liang[1], Yadveer S. Grewal [1], Andri Wardiana[1]
Muhammad J.A. Shiddiky[1], Robert A. Gardiner[4], Hemamali Samaratunga[4], Maher K. Gandhi [5],
Rodney J. Scott[3], Darren Korbie[1] & Matt Trau[1,2]

Epigenetic reprogramming in cancer genomes creates a distinct methylation landscape encompassing clustered methylation at regulatory regions separated by large intergenic tracks of hypomethylated regions. This methylation landscape that we referred to as Methylscape is displayed by most cancer types, thus may serve as a universal cancer bio-marker. To-date most research has focused on the biological consequences of DNA Methylscape changes whereas its impact on DNA physicochemical properties remains unexplored. Herein, we examine the effect of levels and genomic distribution of methylcy-tosines on the physicochemical properties of DNA to detect the Methylscape biomarker. We find that DNA polymeric behaviour is strongly affected by differential patterning of methyl-cytosine, leading to fundamental differences in DNA solvation and DNA-gold affinity between cancerous and normal genomes. We exploit these Methylscape differences to develop simple, highly sensitive and selective electrochemical or colorimetric one-step assays for the detection of cancer. These assays are quick, i.e., analysis time ≤10 minutes, and require minimal sample preparation and small DNA input.

[1] Center for Personalised Nanomedicine, Australian Institute for Bioengineering and Nanotechnology (AIBN), Corner of College and Cooper Roads (Bldg 75), The University of Queensland, Brisbane, QLD 4072, Australia. [2] School of Chemistry and Molecular Biosciences, The University of Queensland, Brisbane, QLD 4072, Australia. [3] School of Biomedical Science and Pharmacy, The University of Newcastle, Callaghan, NSW 2308, Australia. [4] School of Medicine, The University of Queensland, Brisbane, QLD 4072, Australia. [5] Diamantina Institute, The University of Queensland, Brisbane, QLD 4072, Australia. These authors contributed equally: Abu Ali Ibn Sina, Laura G. Carrascosa. Correspondence and requests for materials should be addressed to L.G.C. (email: laura.carrascosa@mater.uq.edu.au) or to M.T. (email: m.trau@uq.edu.au)

DNA methylation is a key epigenetic change involving addition of a methyl group to cytosine nucleotides, and this modification is used by living systems to control genes and their genetic programs[1–3]. The unique levels and pattern of cytosine methylation across the entire genome defines the epigenetic state of the cell, reflects the tissue of origin and when epigenetic reprogramming occurs, it leads to fundamental changes in cell biology which may trigger the onset of diseases[1,2]. An example of this is the transition of cells from a healthy state to malignant neoplasms during cancer[3]. Epigenetic reprogramming in cancer represents a unique methylation landscape involving the net loss of global DNA methylation together with a concomitant increase in the levels of methylcytosines at regions often involved in regulatory roles (e.g., promoter regions), wherein CpG sites are abundant and clustered within a short span[3]. Given the versatile nature of cancer leaving different biomarkers for different cancer types, epigenetically reprogrammed methylation landscape (i.e., Methylscape) is found to be a common feature exhibited by most cancer types and therefore can serve as a universal cancer biomarker. However, there is no appropriate platform to detect this "Methylscape" biomarker which could significantly improve the current strategies for cancer diagnosis, stratification, prognosis and responses to therapy.

DNA is one of the best-known naturally occurring organic polymers in nature and recent studies have found that methylation could impact many physicochemical properties of DNA polymer in solution including DNA structure[4,5], flexibility[6–8] and three dimensional conformation[9–11]. This was mainly attributed to the hydrophobic nature and larger size of individual methylcytosines in comparison to the regular cytosine[12]. However, these studies used model DNA systems and therefore broad changes in the physicochemical properties of DNA polymer occurring at the whole-genome level during cancer epigenetic reprogramming largely remained unknown. Advancement of this knowledge could therefore open new opportunities to precisely detect the "Methylscape" biomarker of cancer genomes by analysing their physicochemical properties alone.

Herein, we report a consequence of genome-wide epigenetic reprogramming induced by cancer, which has been overlooked to date: that the key physicochemical properties of purified genomic DNA are fundamentally different between normal and epigenetically reprogrammed cancer genomes and thereby enable us to develop proper platforms for detecting Methylscape biomarker. We find that the genomic DNA derived from normal cells shows greater tendency towards aggregation in aqueous solutions than genomic DNA derived from cancer cells. This appears to be caused by the hydrophobic properties of methylcytosines leading to different DNA polymer conformations in solution, depending on their levels, and particularly, on their patterning —whether they are evenly distributed or enriched in clusters across the genome. Similar patterning effects on polymer solvation are well-known in polymer chemistry. For example, copolymers with block or clustered distributions of their monomers exhibit widely different physicochemical properties than copolymers with a random or even distribution[13].

We also find that the different solvation properties of cancer and normal epigenomes significantly influence their affinity towards bare metal surfaces, such as gold. Although the DNA-gold interaction is highly sequence-dependent, and some evidence suggests that methylcytosines have higher affinity towards bare gold than regular cytosines[14,15], the physical effect of methylation on DNA-gold interaction has not been characterized. In this study, we finely characterize the affinity of genomic DNA towards bare gold in terms of their methylation level and patterning across the genome. We find that in addition to the solvation properties, this interaction is also modulated by different

affinity of methylcytosines and cytosines, and as a function of their clustered or dispersed patterning (i.e. methylation landscape) across the genome, which in turn, can determine the clinicpathological state of the DNA. Using electrochemical and colorimetric techniques, we develop extremely simple, label free and naked eye platforms that can finely detect Methylscape biomarker from cancer genomes based on the level of gDNA adsorption on planar and colloidal gold surfaces respectively. We test our approaches on a large cohort of over 200 human samples (i.e. genomic DNAs extracted from cell-lines, tissues and plasma) representing various cancer types. Our strategies display high simplicity, avoiding any sensor surface functionalization, DNA pre-processing (i.e., no enzymatic or chemical treatment) and amplification routines (i.e. PCR) of current methylation based cancer detection methods. These methods therefore show great promise for translation into an advanced diagnostic scaffold for the rapid detection of cancer within clinical settings.

## Results

**In solution and surface-based self-assembly of epigenomes.** The main hypothesis of this work is that different methylation landscape of normal and cancerous epigenomes may impact their physicochemical and self-assembly properties in aqueous solutions, and as they interact with solid surfaces (Fig. 1a). To investigate this hypothesis, we visualized (using transmission electron microscopy (TEM)) the purified genomic DNA extracted from normal and malignant prostate tissue isolated from a healthy individual and a metastatic cancer patient respectively. Initial observations of DNA isolated from the cancerous sample put in evidence a uniform coating across the surface, as compared to the normal DNA sample, which showed tendency to create larger aggregates (Fig. 1a, see Supplementary Figure 1 for more images). Digital image analysis showed that the size of aggregates in DNA derived from normal prostate tissue DNA is approximately 8298 $nm^2$, with some of the individual aggregates reaching up to micron sizes (~8 $\mu m^2$). In contrast, the average size of aggregates in cancer tissue DNA is 1540 $nm^2$ with most of them within the nanometre size (see Supplementary Figure 2 for details).

To further investigate the methylation-dependent self-assembly properties of epigenomes in solution, we visualized an additional set of control samples with defined DNA methylation characteristics: (i) a DNA derived from the BT474 breast cancer cell line, which has ~43% global methylation levels (see methods section for calculation details); (ii) a fully unmethylated epigenome generated by whole-genome amplification (WGA) of the BT474 DNA, which is a process that erases all methylation marks but preserves the genetic sequence; and (iii) a commercially available 100% methylated DNA sample (M-Jurkat), which has been enzymatically manipulated to have all CpG sites methylated. The TEM Images depicted in Fig. 1b shows that the unmethylated WGA DNA coated the surface in a uniform manner, but as the sample becomes methylated, nanometer-sized domains begin to emerge (Fig. 1b, WGA verses BT474) with the 100% methylated sample exhibiting large, micron-sized aggregates. The average size of aggregates for BT474 and 100% methylated Jurkat DNA were approximately 3151 $nm^2$ and 8319 $nm^2$ respectively (see Supplementary Figure 4). Interestingly, the TEM image of the fully methylated DNA largely resembles that of normal genome (Fig. 1a) and this could be due to the fact that normal genomes also feature large levels of global methylation. Overall, the TEM data suggest a trend towards increased aggregation with increased global methylation content of DNA epigenomes in solution (Fig. 1b, right). Previous studies suggest that the methyl group is highly hydrophobic, and hydrophobic forces are indeed vastly

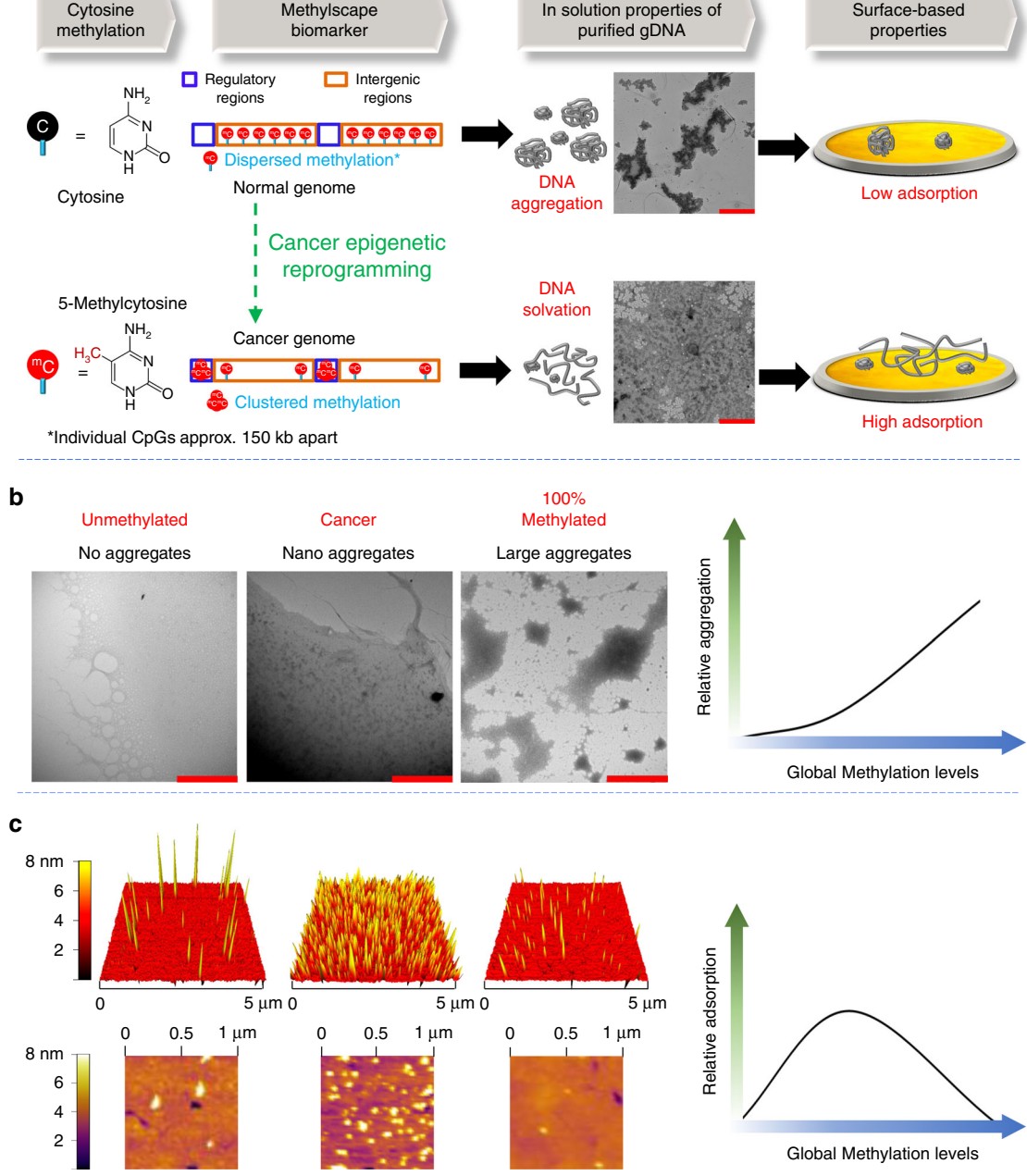

**Fig. 1** Epigenetic reprogramming modulates the physicochemical properties of genomic DNA. **a** Scheme: DNA from normal cells contain large levels of dispersed methylcytosines across the genome but DNA from cancer cells are hypomethylated at these sites and tend to form cluster of methylcytosines into CpG rich regulatory regions. This distinct methylation landscape leads to different solvation properties in solution, which in turn modulates their adsorption towards gold surface. Inset: TEM image showing the different solvation of DNA genome derived from the prostate tissue of a cancer patient and a healthy individual. **b** TEM image showing the different solvation of DNA based on their different methylation status. i) fully unmethylated WGA DNA ii) moderately methylated DNA from BT474 cancer cells iii) 100% CpG methylated Jurkat DNA. The solvation trend of DNA with increasing methylation levels is shown in right side. Scale bars are 2000 nm for all the TEM images. **c** AFM image showing the interaction behaviour of genomic DNAs with the gold surface based on their different methylation status. WGA and 100% methylated jurkat DNA shows very low adsorption, whereas BT474 DNA shows very high adsorption. The surface adsorption trend of genomic DNA with increasing global methylation levels is shown in right side

involved in aggregation processes of polymers[16]. Moreover, hydrophobic driven methylation-dependent conformational changes of DNA have already been reported in the literature[12]. Thus, we assume that the presence of very high methylation levels in the fully methylated and normal DNAs possibly make the DNA polymer highly hydrophobic in nature and thus favour the aggregation process in solution. Additional TEM experiments with these DNA (WGA, BT474 and 100% Methylated Jurkat DNA) were also performed, and technical replicates of different

samples of the same DNA (analysed in different days) continued to display the same surface-interaction effects, suggesting a consistent phenomenon unrelated to sample manipulation or imaging (See Supplementary Figure 3 for additional TEM images).

Following these observations, we envisioned that the distinct nanometer-sized morphologies of cancer vs normal genomes would have an impact on DNA adsorption processes onto metal surfaces such as gold. While the relative gold-affinities of

canonical DNA bases is well known[17–19], the effect of methylcytosines on DNA adsorption has been widely overlooked owing to generalized use of short synthetic oligos or amplified DNA samples that have lost their methylation information during the amplification process.

To evaluate the effect of methylation on DNA-gold adsorption, we first adsorbed the same three DNAs (i.e. WGA, BT474 and 100% Methylated Jurkat) onto ultraflat gold substrates (Roughness ($R$) = 259.4 pm) and visualized them under Atomic Force Microscope (AFM). To date, there are few experiments involving DNA adsorbed onto gold surfaces, and in most cases, they involve the use of DNA chemically anchored onto gold by one end[20,21]. Alternatively they employed DNA sequences (e.g., short oligos or fragments generated by PCR amplification reactions)[22,23], which did not incorporate any methylation information. Hence, to the best of our knowledge, there are no AFM reports involving adsorption of intact genomic DNA onto the gold surfaces, nor comparing full genomes with different methylation landscapes.

As shown in the AFM images of Fig. 1c, unmethylated WGA DNA gave a scattered low-adsorption profile. This observation is in-line with previous reports of unmethylated or amplified double-stranded DNA, which also displayed very low adsorption competence towards gold surfaces[24]. We also observed that, as the genomic DNA became methylated (i.e., BT474 DNA), the surface-adsorption process became significantly favoured, resulting in high saturation of the gold surface (Fig. 1c). However, when the DNA was highly methylated (i.e., 100% methylated DNA), minimal attachment of the epigenome to the gold surface occurred. This suggests that the surface-adsorption kinetics of fully methylated sample is unfavourable presumably due to the prior self-assembly of DNA in solution which formed large aggregates. These large aggregates would therefore limit the available surface-area of the sample and block the potential sample-surface interaction sites. ImageJ analysis of AFM images of these DNA shows that the approximate area coverage for WGA and 100% Methylated Jurkat DNA is 2.80 and 1.34%, respectively (Supplementary Figure 5). In contrast, the approximate area coverage for BT474 DNA is 21.24%, suggesting that there is a parabolic adsorption trend of DNA towards gold surface with increasing methylation levels (Fig. 1c, right).

**Methylcytosines enhance gold-adsorption of methylated DNA.** To obtain greater insight, we used electrochemical assays to quantitatively assess the adsorption levels of DNA with various methylation levels (Fig. 2). The electrochemical assay involved the direct adsorption of 5 μL of purified DNA (10 ng/μL concentration in SSC5X buffer at neutral pH) onto gold electrodes for 10 min. Subsequently, the adsorption competence was measured by Differential Pulse Voltammetry (DPV) in presence of the [Fe(CN)$_6$]$^{3-/4-}$ redox system (Fig. 2a, see methods section for details). Upon adsorption of DNA on gold electrodes, [Fe(CN)$_6$]$^{3-/4-}$ redox system generates a Faradaic current signal, which is proportionally lower than the bare electrode signals[25–27] (i.e., the greater the DNA adsorption is, the larger the relative current signal difference, %$i_r$, with respect to the original baseline. We have previously used this redox system to quantify gold-adsorbed DNA[25,26] and RNA[28,29] with excellent precision, and to discriminate between short DNA sequences with single-base differences under optimized conditions.

Using this approach, we tested a collection of genomic DNAs with the following (i) no methylation (ii) significant hypomethylation (iii) moderate methylation, (iv) large CpG Methylation (v), and 100% CpG methylation. The unmethylated and fully methylated DNA genomes were the same DNAs (i.e., WGA, and 100% methylated Jurkat) that we tested in our previous TEM

and AFM experiments. The hypomethylated DNA used for this experiments was gDNA derived from Jurkat cancer cells grown in the presence of 5-azacytidine—a demethylation drug that generates DNA with an average of <30% global methylation. For moderately methylated DNA, we used BT474 and Jurkat cancer cell derived DNA which have 43 and 36% global methylation respectively and for largely methylated DNA, we used Human Mammalian Epithelial cell (HMEC) line DNA representing the normal phenotype (global methylation = 64%). In agreement with our previous AFM data, the BT474 and Jurkat DNA provided significantly higher adsorption levels as reflected by ~20–40% larger relative current than the unmethylated, normal and the 100% methylated genomic DNA (Fig. 2b, %$i_r^{BT474}$ = 41.58 ± 0.87, %$i_r^{Jurkat}$ = 49.24 ± 1.17 vs. %$i_r^{WGA}$ = 10.58 ± 1.09, %$i_r^{HMEC}$ = 18.18 ± 0.81, %$i_r^{M-Jurkat}$ = 11.28 ± 0.49; where ± represents the standard deviation of the measurements, for optimization experiments, see Supplementary Figure 6). Largely methylated HMEC DNA and the fully methylated DNA, led to poor adsorption, and so did demethylated aza-Jurkat genomes, whose levels were slightly larger than the unmethylated WGA version, but still far lower than the cancer derived BT474 and Jurkat DNA (Fig. 2b). This data also indicate that the adsorption of genomic DNA towards gold surface varies with the number of methylcytosines present in the genome. Within this sample data set, adsorption appears to display a parabolic trend with the increase of global methylation percentage (Fig. 2f); with the highest values for methylation percentages typical of cancer-derived DNA

To better ascertain the role of methylcytosines in DNA-gold adsorption, we performed another set of experiments with short DNA fragments and individual nucleotides with different methylation status. In a first suite of experiments, we compared the adsorption behaviour of 1 ng/μL of ds-DNA fragment (140 bp long, see "Methods" section for the detail sample preparation) encompassing a cluster of either eight methylcytosines (M-DNA) or cytosines (UM-DNA) at neutral pH for 20 min. These two DNAs exhibited markedly different adsorption trend (Fig. 2c) with the M-DNA showing 20% larger relative current signal difference than UM-DNA (i.e., %$i_r^{M-DNA}$ = 37.47 ± .2.51 vs %$i_r^{UM-DNA}$ 17.15 ± 2.50). This outcome suggests that the small methylated DNA fragments have larger gold-DNA adsorption than the unmethylated fragments. This could be due to a higher affinity of methylcytosine towards gold in comparison to the unmethylated cytosine nucleotide. To confirm this point, we performed a similar experiment with 1 μM solution of methylated (M-dCTP) and unmethylated (dCTP) individual cytosine nucleotides. A significantly higher adsorption was also observed for M-dCTP (Fig. 2d) in this case (i.e., %$i_r^{M-dCTP}$ = 19.45 ± 1.45 vs %$i_r^{dCTP}$ = 12.01 ± 0.78). These data indicate that methylation can modulate DNA adsorption onto gold surfaces in a dynamic way, where adsorption of small fragments and individual nucleotides is enhanced by the presence of methylcytosines. However, in case of whole genomes, methylation appears to only favour adsorption until it reaches a certain methylation value; and once a methylation maxima is exceeded, self-assembly of DNA in solution no longer favours the epigenome-surface interactions. This is presumably due to the formation of large aggregates, as noted in Fig. 1, which are more likely to appear for highly methylated DNA samples, but not for cancer-derived DNA.

To evaluate this hypothesis and further investigate the reason for this methylation maxima self-assembly trend, we designed a time-point whole genome methylation-dependent adsorption experiment. We treated the unmethylated DNA (WGA) with the *M.SssI CpG methyltransferase* which attaches methyl groups to cytosines in CpG dinucleotides. By incubating this unmethylated genomic template with *M.SssI* for increasing time periods,

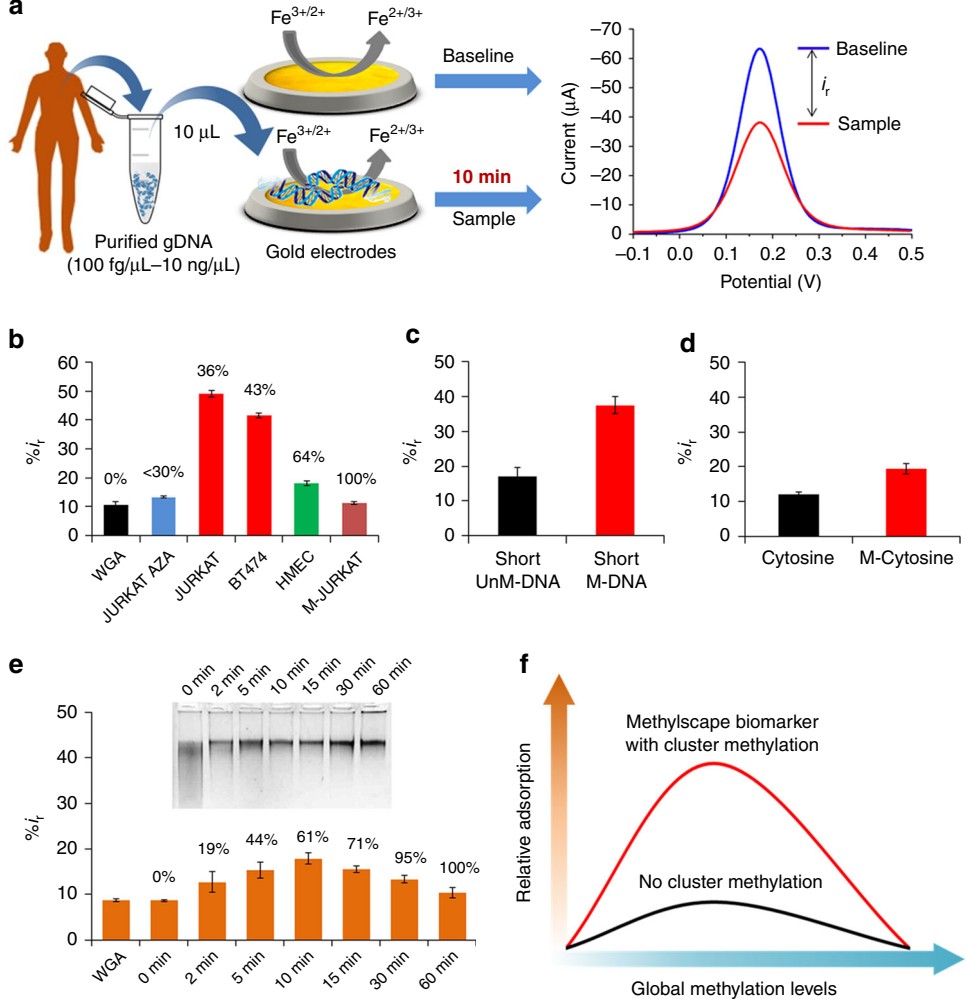

**Fig. 2** Role of methylation on DNA-gold affinity. **a** Methodological approach for the electrochemical quantification of DNA adsorption on gold electrodes. First, the DPV current from the bare gold electrode was measured to get the baseline signal. The purified DNA extracted from cell line, tissue or plasma samples were then adsorbed onto the gold electrodes and the DPV current was measured to get the sample signal. The difference between the baseline and the sample signal is the $i_r$ value, which is normalized to %$i_r$ for better understanding. The %$i_r$ for a given DNA sample directly correlates with the adsorption level of DNA on gold electrodes. **b** Relative current (%$i_r$) mean values for the unmethylated WGA DNA (black), Aza treated demethylated Jurkat DNA (Blue), gDNA from BT474 and Jurkat cancer cells (red), HMEC gDNA from primary mammary cells (green) and 100% methylated gDNA from Jurkat cells (Brown). **c** Bar graph of relative current (%$i_r$) mean values for the adsorption of 140 bp methylated (red) and unmethylated (black) DNA fragments. **d** Bar graph of relative current (%$i_r$) mean values for the adsorption of individual cytosines (black) and methylcytosines (red) nucleotides. **e** Relative current (%$i_r$) mean values for various genomic DNAs prepared from WGA DNA by enzymatic reaction at different time points. Sample methylation levels are provided above each bar. Inset: electrophoresis gel of the enzymatically methylated DNA samples digested with methylation sensitive HpaII restriction enzyme. **f** Effect of cluster methylation towards adsorption. Each data point for **b**–**e** represents the average of three separate trials, and error bars represent the standard deviation of measurements (%RSD = < 5% for $n = 3$)

we generated a series of DNA genomes with increasing methylation content. The outcome of this enzymatic treatment was confirmed by digestion using methylation sensitive HpaII restriction enzyme. ImageJ analysis of gel picture for HpaII enzyme digestion experiment (Fig. 2e, inset) allowed rough estimation of the methylation levels for each sample treated with M.SssI at a given time-point (See Supplementary Figure 7 for details). The results of this experiment indicated that increasing methylation levels led to increased epigenome-surface interaction as measured by electrochemical DPV assays until a methylation threshold was reached. Beyond this threshold, sample adsorption onto the gold surface was compromised and adsorption levels began to decrease (Fig. 2e). Interestingly, it was not possible to recapitulate the overall adsorption value generated by the

cancerous BT474, or Jurkat epigenome samples. For example, the BT474 sample with overall methylation levels in the range of 43% led to adsorption levels approximately two and half-times larger than the maxima achieved with the M.SssI samples (i.e., %$i_r^{BT474}$ = 41.58 ± 0.87 vs %$i_r^{10mins}$ = 17.9 ± 1.2). This observation suggest that although BT474 DNA have similar methylation levels as the DNAs obtained from 5-10 min M.SssI samples, it would portray a specific methylation pattern that favour DNA adsorption significantly, to a degree unseen in any other tested DNA sample, regardless of their global methylation content. We hypothesize that this would be caused by the presence of "Methylscape" biomarker (i.e. regions with high levels of clustered methylation separated by large intergenic tracks of unmethylated regions) in cancer genomes where the distribution

of methylcytosine is significantly different than the DNAs methylated with the *M.SsI* enzyme. This is because this enzyme attaches methyl group across the genome in random fashion rather than in clustered regions. Therefore, this Methylscape biomarker, which is typical of cancer DNAs and not present in DNAs from healthy individuals, would drive a unique self-assembly process, and regardless of their global methylation content, creates a distinctive adsorption footprint (As shown in Fig. 2f) that can be used to infer their clinicopathological state.

**Methylscape can discriminate normal and cancerous samples**. From the above experiments, we hypothesized that the cluster methylation also called regional hypermethylation present in cancer genomes represented an ideal configuration for max-imising epigenome-surface adsorption. A corollary of this pro-position is that the unique self-assembly process of cancerous epigenomes, due to their methylation landscape distribution, could be exploited to detect "Methylscape" biomarker using biosensing applications.

To investigate the possibility of developing a simple method for cancer detection based on the different physicochemical proper-ties of DNA, we used our electrochemical DPV assay to analyze various epigenomes extracted from breast (BT474, MCF7 and T47D), prostate (LNCap), lung (H1975) and colorectal (HCT116) cancer cell lines, and compared them to DNA isolated from healthy breast (HMEC) or prostate (PrEC) cells. Notably, DNA genomes from breast cancer or prostate cancer cells provided ~2.5-fold higher relative current than their respective normal breast (HMEC) and prostate (PrEC) cell lines, as did the other epigenomes isolated from lung (H1975) and colorectal (HCT116) cancer cells (Fig. 3a).

To determine the applicability of our approach for cancer detection in clinical specimens, we analysed 72 epigenomes extracted from patient tumour tissues of different cancer types (54 ER+ breast, 8 prostate, and 10 lymphoma cancer tissues), and compared their adsorption levels to 31 epigenomes extracted from matching tissues types of healthy individuals (19 normal breast, 10 normal prostate and 2 normal lymph node tissues). Figure 3b shows the individual box plot representing relative DPV current values for the gDNA samples extracted from breast (ER+), prostate and follicular lymphoma cancer tissues verses matching tissues from healthy individuals (See Supplementary Figure 14, 15 and 16 for original DPVs and Supplementary Table 6–9 for clinical information). We observed significant differences in gDNA adsorption levels between normal and cancer samples when they are compared by tissue type (Fig. 3b) or when all cancer types were combined (Fig. 3c). The combined box plot (Fig. 3c) for the adsorption experiments of all three types of cancer and normal samples shows that 75% of cancer samples have a relative DPV current ($\%i_r$) value of more than 25 units, whereas 75% of gDNA derived from normal tissues provide relative current values lower than 20 units. Statistical significance was determined by pairwise comparisons between normal and cancer samples using Student's *t*-test for each of the box plots. *P* value of the *t* test (Supplementary Table 1) clearly shows that the normal and cancer samples are significantly different with 95% confidence. Moreover, the ROC curve (Fig. 3c) for the range of tissue samples tested shows high-specificity for cancer detection (AUC = 0.909). Statistical diagnostic efficacy test at cut-off value $\%i_r = 20$ shows that our method has high accuracy (89.32%) with high positive (PPV) and negative (NPV) predictive values (Table-Fig. 3c, PPV = 91.78%, NPV = 83.33%, see more details at Supplementary Table 2). Notably, most of the samples used in this study were isolated from patient and normal individuals with the age above 40 years (See Supplementary Table 6–9). For the

breast and prostate cancer samples, comparison with the normal samples was performed among individuals of the same gender. This way, our analysis is not biased by gender or age associated DNA methylation changes among individual's DNA. Finally, to validate the methylation dependent adsorption changes of genomic DNA, we have quantified the global methylation levels of some of our patient and normal DNA samples (See Method section for details). As shown in supplementary table 6–9, most of the patient DNA samples have moderate or low methylation levels in the range of 30–50 percent while the normal DNA samples have higher level of global methylation on average of 50–75 percent. These data validates our initial hypothesis that the adsorption of genomic DNA onto the gold surface is significantly modulated by the global methylation levels and patterning that defines the proposed Methylscape biomarker.

**Methylscape applications using circulating free DNA**. To develop a non-invasive assay, we next sought to analyse circu-lating free DNA (cfDNA) isolated from plasma samples of 100 breast (ER+) and colorectal cancer patients and compared their adsorption with normal plasma cfDNA derived from 45 healthy individuals (See Supplementary Table 10, 11 and 12 for clinical information). In this case, only 5 pg (concentration: 1 pg/μL) of plasma derived patient cfDNA were sufficient, and adsorption was carried out for 10 minutes, followed by electrochemical measurements. Similar to previous experiments, the cfDNA extracted from cancer patients showed higher relative current ($\%i_r$) than cfDNA samples derived from the plasma of healthy individuals (Fig. 3d, $\%i_r = > 35$ for 75% of cancer samples and $\%i_r = < 35$ for 75% of normal samples). The *P* values clearly indicate that the normal and cancer samples are significantly different with 95% confidence (see Supplementary Table 1). The area under the ROC curve (AUC = 0.887) is also similar to that observed for tissue samples. Statistical diagnostic efficacy test at cutoff value $\%i_r = 35.7$ shows that our method has high accuracy (83.45%) with high positive and negative predictive values (Table-Fig. 3d, PPV = 91.30%, NPV = 69.81%, see more details at Sup-plementary Table 3). A separate study with only breast and col-orectal samples also showed very high sensitivity and specificity (see Supplementary Figures 10 and 11 for details). To eliminate any bias towards gender and age of the patients, we have also performed the age and gender matched analysis using the cfDNA samples. We compared 13 samples from 40–60 years old breast cancer female patients with 13 samples of healthy individual within the same gender and age range. The box-plot presented in supplementary figure 12 shows that the adsorption value of patient samples are clearly distinguishable from the normal sample (Area under the ROC curve is 0.923). We have also compared samples from 50 colorectal cancer patients with 19 samples from healthy individuals within the age range of 50-80 years. As shown in Supplementary Figure 13, cfDNA from col-orectal cancer patients provided higher gold adsorption in com-parison to the normal samples (Area under the ROC curve is 0.842). This data clearly indicates that our approach is not biased by gender and age related methylation changes in the genome. Notably, the use of cfDNA for detection allows ultra-low sample input requirements. Optimization of our assay conditions enabled detection from as low as 500 fg of purified cfDNA input (See Supplementary Figure 9). To further test the sensitivity of out method, we designed another experiment which involved spiking of different proportion of a cluster methylated DNA template into the normal plasma derived cfDNA solution. This experiment is important because it is noted in the literature that cfDNA variant allele frequency (VAF) is below 10% and in some cases even below 1% in the plasma samples of cancer patient. Thus to

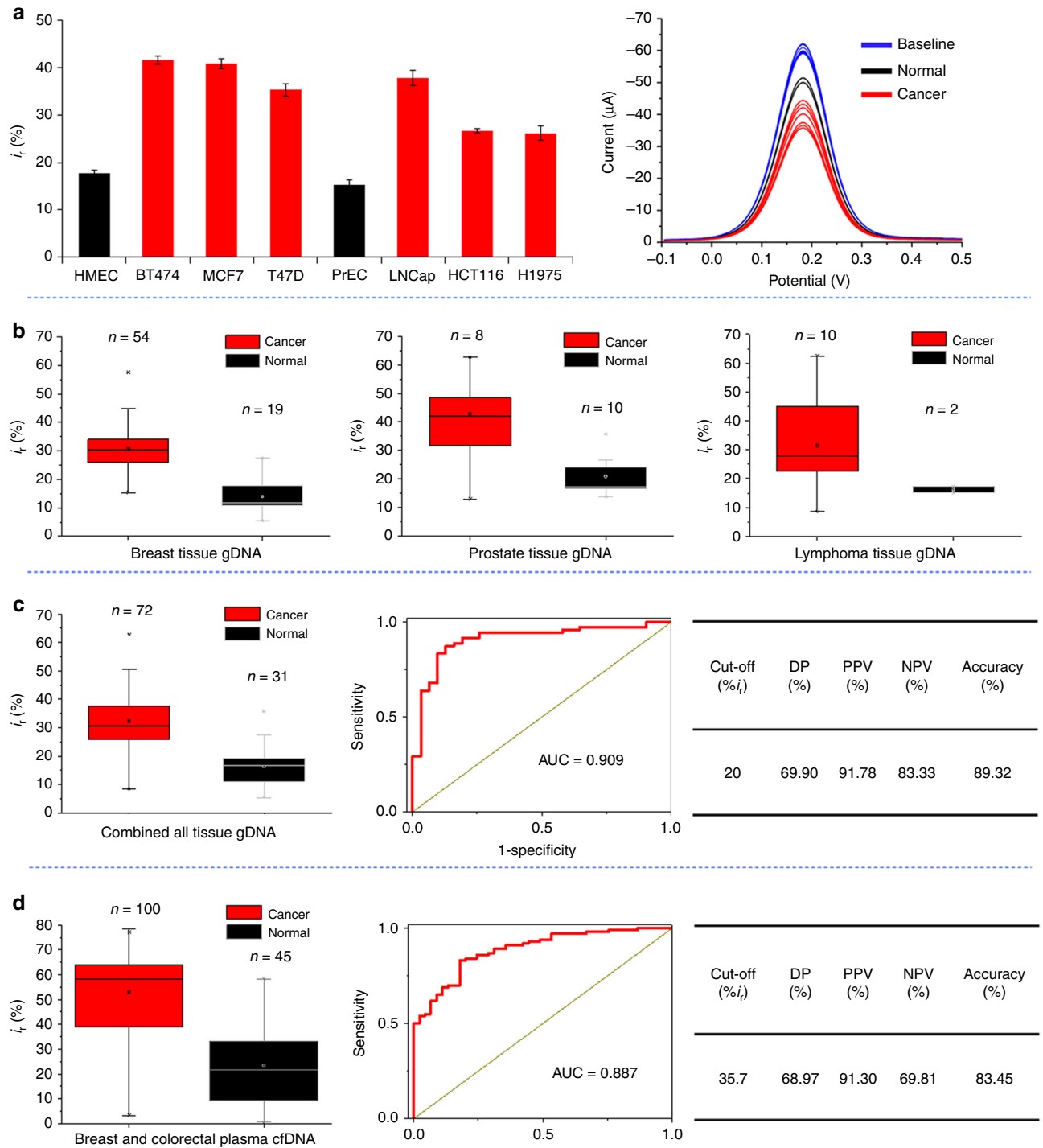

**Fig. 3** Practical application of Methylscape approach. **a** Differential affinity/adsorption of genomic DNAs as a function of their cancer and normal origin. Bars represent the relative current (%$i_r$) mean values for DNA genomes derived from various normal (Black bars: Human Mammalian Epithelial Cell (HMEC), Prostate Epithelial cell (PrEC)) and cancer cell lines (Red bars: Breast cancer: BT474, MCF7, T47D, LNCap prostate cancer, HCT116 colorectal cancer and H-1975 lung cancer). Each data point represents the average of three separate trials, and error bars represent the standard deviation of measurements (%RSD = < 5% for n = 3). Right panel: Corresponding DPV graphs for all cell line DNAs and their respective baseline. **b** Individual box plot showing mean relative current (%$i_r$) values generated by electrochemical detection of genomic DNAs extracted from normal and cancer tissues of various cancer types (breast, prostate and lymph node). **c** The combined data for all the tissue samples derived from 31 healthy individuals and 72 cancer patients, Right: the ROC analysis and diagnostic test evaluation shows the Disease Prevalence (DP), positive predictive values (PPV), negative predictive values (NPV) and accuracy of the method. **d** Box plot showing relative current (%$i_r$) mean values generated by electrochemical detection for genomic DNAs extracted from plasma samples derived from 45 healthy individuals and 100 breast cancer patients, Right: the ROC analysis and diagnostic test evaluation shows the Disease Prevalence (DP), positive predictive values (PPV), negative predictive values (NPV) and accuracy of the method. In the box and whisker plots, the middle lines of the boxes represent the median (50th percentile) and the terminal line of the boxes represents the 25th to 75th percentile. The whiskers represent the lowest and the highest value

address this question, we thought to see whether our method is sensitive enough to detect very low percentage of cluster methylated template DNA in presence of large numbers of normal cfDNA sequences. We therefore used our short and cluster methylated template DNA (used in experiment for Fig. 2c) and spiked this DNA in normal cfDNA solution at different proportion (0%, 0.1, 1, 2.5, 5, 10). As shown in the supplementary figure 18, the relative adsorption of cfDNA increased with the increase of methylated template DNA in the solution and can detect low loading of methylated DNA fragments, but it was not possible to clearly detect levels in the range of 1% or less. This limit of detection may not be sufficient for very low levels of tumour cfDNA and with the current version of the method; we may not be able to detect cancer on a very early stage, as patients may have cancer DNA copies, usually expressed as mutant variant allele frequency (VAF), in levels below 1% in plasma.

**Naked eye detection of cancer using AuNPs.** To assess the efficacy of our approach in detecting Methylscape biomarker using colloidal gold, we utilized salt-induced AuNP aggregation system[30–34]. For DNA analysis, salt-induced aggregation of AuNP is arguably the most suitable format for detection, due to excellent sensitivity, reproducibility and ease of performance[35]. In this approach, the AuNP aggregates upon addition of salt unless they are protected by previously adsorbed DNA molecules. This aggregation process can be detected by naked eye as a visual colour change of the AuNP solution from reddish to blue due to the red shift of the localized nanoparticles' surface plasmon band[32,35]. We therefore incubated 50 ng of purified DNA with AuNPs for 5 min, followed by the addition of salt (SSC 5X) to induce aggregation (see "Methods" section for details). Measurement of the spectral shift generated upon salt addition, showed approximately 6.5 units higher relative absorbance ($A_{658/520}$) for BT474 DNA compared to the unmethylated WGA (i. e., $A_{658/520}^{BT474} = 7.23 \pm 0.85$, vs. $A_{658/520}^{WGA} = 0.74 \pm 0.04$, Fig. 4b). The assay was also sensitive enough to identify 5-azacytidine treated Jurkat DNA (Fig. 4b). Of note, any of the genomic DNAs artificially methylated by the *M.SssI CpG methyltransferase* and also the 100% methylated Jurkat DNA showed very poor ability to stabilize the AuNPs in solution (Supplementary Figure 19)—an observation in-line with our previous data and concordant with our methylation maxima self-assembly model.

To further investigate how the self-assembly of different epigenomes affects their interactions with AuNPs, we used TEM to visualize DNA-AuNPs interactions before addition of salt (Fig. 4c, Supplementary Figure 20). As shown in Figure c (AuNP alone) and d (AuNP and WGA), the presence of unmethylated DNA (WGA) had minimal effects on the dispersion pattern of AuNP, suggesting limited interaction of unmethylated DNA with colloidal gold. In contrast, the presence of moderately methylated DNA from BT474 cells (Fig. 4e) favoured a dispersed distribution of AuNPs. We hypothesize that the larger ability of BT474 DNA to stabilize AuNPs is due to the particular methylation landscape and high affinity of this type of DNA towards gold surfaces. Interestingly, fully methylated Jurkat DNA (Fig. 4f) appeared to interact strongly with AuNPs whereas in the case of flat gold surface, it poorly interacted (Fig. 1c, AFM). We hypothesize that the difference in interaction between AuNPs and solid surfaces is associated to the ability of colloidal gold particles to move around DNA aggregates. This would allow them to perfuse the methylated-DNA aggregate to find methylcytosine-rich spots for interaction. However, because of the large numbers of AuNPs interacting with fully methylated DNA, this system displays the AuNPs in close proximity to each other and ultimately collapsed

into large colloidal aggregates upon salt addition —probably by a crosslinking aggregation mechanism. The average DNA-AuNP aggregate size obtained from ImageJ analysis (Supplementary Figure 21) of the TEM images also support our gDNA-AuNP interaction hypothesis.

To assess the clinical utility of the assay, we next tested a cohort of 24 epigenomes isolated from different metastatic cancer types (e.g., ER+ breast, prostate and follicular lymphoma), and compared their adsorption profiles to epigenomes isolated from 24 matching normal tissues (See Supplementary Table 6, 7, 8 and 9 for clinical information). The relative absorbance in Fig. 4g indicates that tumour samples favour AuNP adsorption as compared to DNA from healthy controls. Although the area under the ROC curve (0.761) is comparatively lower than that observed for electrochemistry, statistical diagnostic efficacy test at cutoff value %$i_r = 4$ shows good accuracy (77.08%) with reasonable positive and negative predictive values (Table-Fig. 4g, PPV = 80.95%, NPV = 74.07%, see more details at Supplementary Table 4). However, a larger sample cohort and optimization of experimental conditions might help in further increasing the sensitivity and specificity of this system.

Finally, to test the applicability of naked eye system for non-invasive detection of cancer, we analysed cfDNAs derived from plasma samples of 100 breast and colorectal cancer patients and 45 healthy individuals (See Supplementary Table 10–12 for clinical information). Only 1 pg of cfDNA was required to stabilize the AuNP solution and prevent the salt-induced AuNP aggregation. As shown in the box plot in Fig. 4h, 75% of cancer samples provided a relative absorbance ($A_{520/658}$) value higher than approximately 7 units; whereas, 75% of cfDNA derived from normal plasma showed significantly lower relative absorbance values. The P-values obtained from Student's t-test also confirmed the statistical significance of these data (See Supplementary Table 1). The area under the ROC curve (AUC = 0.785), although slightly lower than that observed for electrochemistry, still shows good sensitivity and specificity. In this system, statistical diagnostic efficacy test at cutoff value %$i_r = 8.7$ provides an accuracy of 73.10% with reasonable positive and negative predictive values (Table-Fig. 4h, PPV = 88.61%, NPV = 54.55%, see more details at Supplementary Table 5). Similar to the electrochemistry experiment, the separate study with only breast and colorectal samples also showed good sensitivity and specificity (see Supplementary Figures 22 and 23 for details). Furthermore, to eliminate any bias towards gender and age of the patients, we have performed the age and gender matched analysis using the nanoparticle based approach. We compared 13 cfDNA samples from 40- to 60-years-old breast cancer female patients with 13 samples of healthy individual within the same gender and age range. The box-plot presented in supplementary figure 24 shows that the adsorption value of patient samples are fairly different from the normal sample (Area under the ROC curve is 0.834). We have also compared samples from 50 colorectal cancer patients with 19 samples from healthy individuals within the age range of 50-80 years. As shown in supplementary figure 25, cfDNA from colorectal cancer patients provided higher gold adsorption in comparison to the normal samples (Area under the ROC curve is 0.719). These data clearly outline that naked eye assay could potentially detect the presence of cancer in a rapid and cost-effective manner.

**Proposed mechanism of detecting Methylscape biomarker.** The work presented here is based on the detection of a global methylation landscape in cancer, which we referred to as Methylscape. The Methylscape in cancer genome involves a change in global methylation levels and patterning in comparison

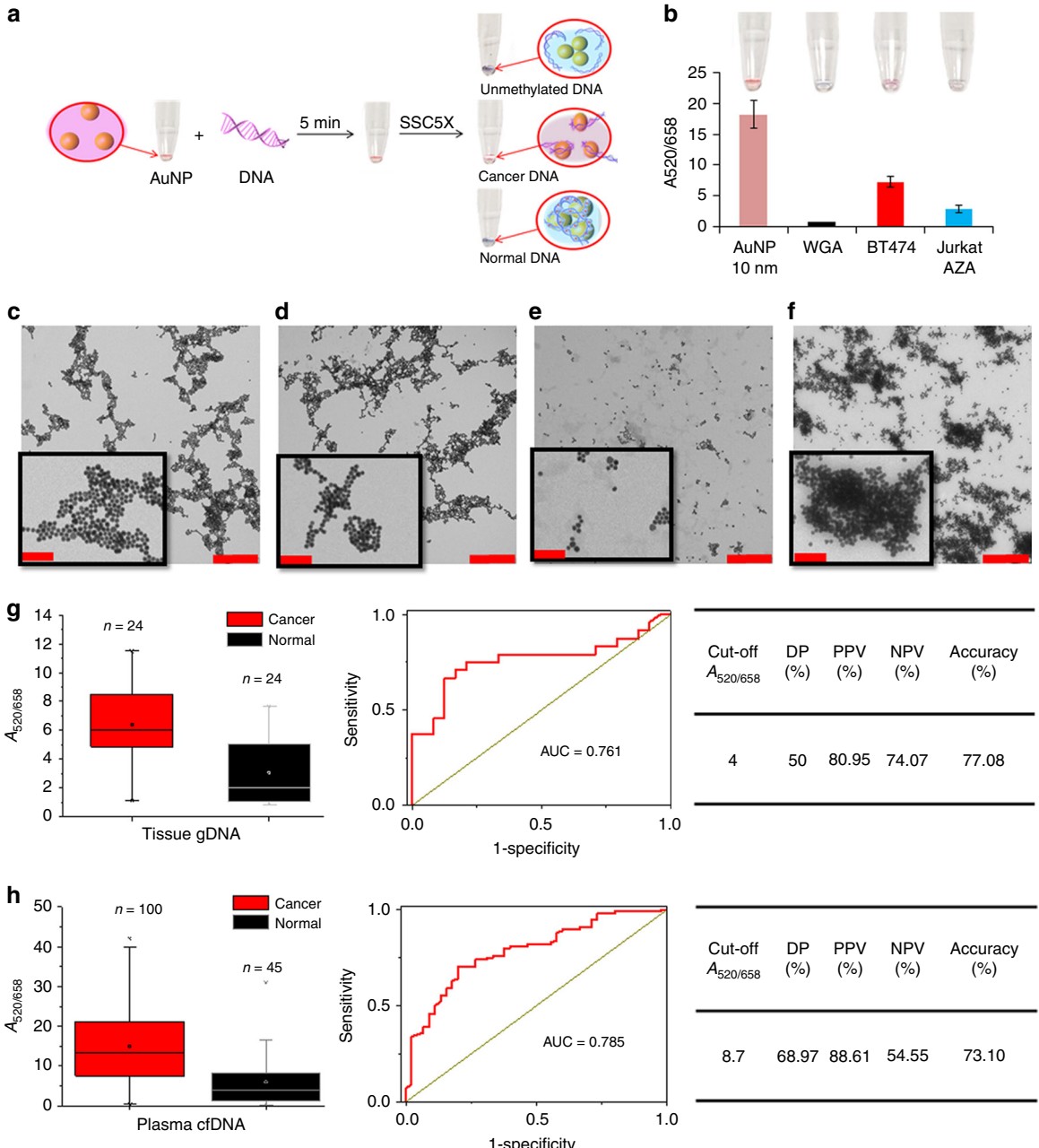

**Fig. 4** Naked eye detection of cancer using AuNPs. **a** Schematic of the assay and proposed mechanism for different DNA types. **b** Mean relative absorbance values $A_{520/658}$ of 10 nm tannic-capped AuNP (pink) and AuNP- gDNA solution for unmethylated WGA (black), BT474 breast cancer cell line (red), and Aza treated Jurkat (light blue). Inset, the representative coloured solution. TEM images of **c** AuNPs alone, **d** AuNP-gDNA solution for unmethylated WGA, **e** AuNP with BT474 cancer DNA and **f** AuNP with 100% methylated Jurkat DNA (no salt was added during sample preparation, scale bars are 500 nm for normal images and 100 nm for the zoomed images in the black boxes). **g** Box plot showing the mean relative absorbance values $A_{520/658}$ of AuNP-gDNA solution for cancer and normal cells extracted from breast, prostate and lymph node tissues, Right: The ROC analysis and diagnostic test evaluation shows the Disease Prevalence (DP), positive predictive values (PPV), negative predictive values (NPV) and accuracy of the method. **h** Box plot showing the mean relative absorbance values $A_{520/658}$ of AuNP-gDNA solution for DNA samples derived from plasma samples of breast and colorectal cancer patients or healthy donors, Right: The ROC analysis and diagnostic test evaluation shows the Disease Prevalence (DP), positive predictive values (PPV), negative predictive values (NPV) and accuracy of the method. In the box and whisker plot, the middle lines of the boxes represent the median (50th percentile) and the terminal line of the boxes represents the 25th to 75th percentile. The whiskers represent the lowest and the highest value

to the Methylscape in normal genome. Genomes from adult normal tissues tend to exhibit overall higher degrees of methylation, which are also quite evenly dispersed (uniform) throughout the genome. In contrast, this distribution changes during the course of cancer as DNA gradually loses methylation across the genome and exhibits more defined methylated areas where methylated sites are clustered within a short span[3]. However, within this averaged trend, there is intrinsic heterogeneity in the

DNA methylation patterns across cells within the tissue particularly in the context of cancer. Despite this heterogeneity, the changes in the cell's DNA Methylation pattern and level during cancer progression are well documented in the literature as a key feature of cancer epigenetics[1,3]. It is this global change in the methylation pattern, and overall levels and distribution that our methodology is able to detect in a simplified way and the data presented in this manuscript provides the foundations for considering this phenomenon as a general biomarker for cancer.

Our data show that the Methylscape biomarker, which represent a unique footprint for cancer genomes, modulate the self-assembly of methylated DNA in solution and during their adsorption towards gold surfaces. However, self-assembly of DNA appears to be a process with dynamic tension, where adsorption of DNA onto a surface is initially enhanced by the presence of methylcytosine until it reaches an adsorption maximum in low-to-moderately methylated epigenomes, but once a methylation maxima threshold is exceeded the self-assembly process makes epigenome-surface interactions unfavourable. Given this, we hypothesize that due to the large tracts of uniformly methylated regions in normal DNA, large number of hydrophobic methyl groups in solution come into proximity with each other and collapse into self-contained nano- and micro-sized domains surrounded by hydrophilic unmethylated regions, whose surface would then have the same properties and adsorption affinity as a fully unmethylated DNA. The empirical data presented in this study supports this theory, and explains why a 100% methylated and heavily methylated normal epigenome have similar surface adsorption properties as a completely unmethylated sample (Figs. 2b and 3a). In the same line, the fact that cancer cells have large tracts of variably demethylated DNA (with a high degree of heterogeneity) with hypermethylated CpG islands are also in agreement with our hypothesis. Despite some degree of variable demethylation across the genome, the reduction in the overall methylation levels compared to normal genomes, would reduce overall hydrophobicity of the DNA colloid and the chances for DNA to collapse into the above-described self-contained nano- and micro-sized domains. This, in turn, would contribute to increase its overall solubility in aqueous solutions and the chances for hyper-methylated CpG islands to be more accessible and exposed for interacting with the gold surface. This model is supported by the empirical data presented in Fig. 2e and supplementary figure 19, which show an increasing affinity of methylated DNA towards gold surfaces until the methylation maxima is reached. However, below the methylation maxima, the presence of Methylscape biomarker (i.e. cluster methylation separated by large hypomethylated regions across the genome) which is uniquely featured in cancer epigenomes starts to dominate the adsorption process. As a result, the adsorption maximum (as shown in Fig. 2f) is significantly increased in case of cancer epigenomes —a trend that was not observed for artificially methylated (MssI enzyme) derived epigenomes with similar global methylation content.

While the interaction of methyl groups and methylcytosines with gold surfaces has previously been considered[14,15,36], the mechanism which drives their increased adsorption affinity towards gold is still unknown. On the basis of the data presented here, we hypothesize that the electron donating properties of the methyl group might enhance the adsorption of methylated cytosines towards gold via an electron-donating mechanism, as methyl groups can donate electrons and increase the $\pi$ electron density of the cytosine ring[37,38]. Moreover, since the cytosine-base can interact with the gold through the pyrimidine ring in a planar manner (i.e., it aligns parallel to the surface)[39] the increased $\pi$ electron density in the ring may increase the chance of metal-cytosine $\pi$ back-bonding—that is, bonding between the

vacant $d$ orbital of the metal and the $\pi$ electron of cytosine. This metal-cytosine back-bonding could possibly increase the intrinsic affinity of methylcytosines towards gold compared to the unmethylated cytosine. Thus, the cluster methylation, which is uniquely over-represented in cancer genomes, could offer an ideal configuration for adsorption due to numerous methylcytosines in close proximity. This proximity could synergistically deliver a stronger force for holding the gDNA onto the planar gold surface proving higher adsorption. Furthermore, hypermethylated CpG rich regions featuring CG-repetitions[40,41], could also enhance DNA gold-adsorption through these sites because they often experience B → Z transitions when become methylated[10,42–44]. Since Z-DNA is not as tightly wrapped as the B-DNA[40], especially in the B-Z junction, DNA bases within this region could display more favourable orientation for gold-interaction.

The data for cfDNA analysis shows that the sensitivity for cfDNA is significantly improved in comparison to the tissue derived genomic DNA. Although the reason for the sensitivity improvement is unclear, we believe this is associated to their relative size, in average of 165 bp. While this length is in principle sufficient for cancer-derived cfDNA to accommodate a highly methylated CpG island, fragments with the pattern commonly seen in normal samples, where methylated sites are very dispersed (on average 1 methylated CpG every 150 bp)[3] would essentially behave as non-methylated. This may possibly reduce the chances for DNA from normal cells to interfere with the interaction of cancer-derived cfDNA with gold, hence increasing the threshold gap to distinguish between them. In addition to this, it has been suggested in the literature that the fragmentation pattern of cfDNA varies as a function of tissue of origin[45–47], and that cancer-derived DNA also tends to be shorter[48,49]. The presence of these smaller fragments, also featuring methylated clusters, may kinetically favour their adsorption compared to larger unmethylated or scarcely methylated cfDNA fragments from normal cells. This would also contribute to increase the adsorption gap difference between normal and cancer-derived cfDNAs significantly, so that a much lower DNA concentration is sufficient to distinguish them.

## Discussion

This study provides fundamental insight about the consequences of epigenetic reprogramming in the physical state of DNA polymer in solution and when it adsorbs onto metal surfaces such as gold. Consequently, the significant difference in the solution and surface based physicochemical properties between cancer and normal DNA has enabled us to detect the proposed Methylscape biomarker in a single step based on a interfacial biosensing strategy (i.e., it only requires direct adsorption of DNA onto the bare gold surface) using electrochemistry and a colloidal gold system. The ability of methylcytosines to enhance the interaction of DNA with colloidal gold particles is a notable discovery. Previous studies have shown that the stiffer ds-DNA has very low adsorption capability in gold nanoparticle systems, in contrast to more uncoiled ss-DNA[35]. While, previous studies only considered the unmethylated ds-DNA, this report demonstrates that methylated ds-DNA can significantly adsorb onto gold nanoparticles due to the higher affinity of methyl-cytosine.

The most remarkable features of our interfacial biosensing strategy are that they can effectively identify the Methylscape biomarker from cancer genomes without extensive sample preparation (e.g., bisulphite or enzyme treatment and PCR amplification) and sensor surface modification—— a laborious process for most bio-sensing techniques. Moreover, our strategy showed large potential for cancer diagnosis as evidenced by the ROC graphs (e.g., AUC = 0.909 for tissue-derived DNA detection with

 

electrochemistry) for gDNAs extracted from cancer and normal tissues representing various organs (i.e., breast, prostate and lymph node). Our approach also enabled non-invasive cancer detection (i.e., a blood test) in 10 min from plasma derived cfDNA samples with excellent specificity (e.g., AUC = 0.887 for cfDNA detection with electrochemistry) and sensitivity (100 fg/μL). We believe that this simple approach (i.e. Methylscape) with the excellent sensitivity and specificity would potentially be a better alternative to the current techniques for cancer detection. However, Methylscape in it's current form is only able to determine the presence of disease and a detailed analysis is required to fully understand the type, stage and disease recurrence.

## Methods

**Materials**. All the cancer cell lines were purchased from ATCC (USA) and cultured in our lab following the standard Protocol. The culture materials such as growth medium (RPMI 1640), fetal bovine serum (FBS) and antibiotics were purchased from Gibco, Life technologies. Human Mammalian Epithelial Cell (HMEC) line DNA was purchased from Science Cell. Aza treated Jurkat demethylated DNA was purchased from New England Biolabs. Breast cancer patient tissue samples were obtained from UK Forever Clinical Trial, UK. Prostate cancer patient tissue samples were collected from Aquesta Pathology, Brisbane, Australia. Follicular Lymphoma nodal tissues were collected from Princess Alexandria Hospital, Brisbane, Australia. Breast cancer plasma samples were collected from UK Forever Clinical Trial, UK. Colorectal cancer patient plasma sample were obtained from Hunter Medical Research Institute, The University of Newcastle, Australia. The relevant ethical approval was obtained from Bellberry Limited, Australia for all tissue and plasma patient samples analysis presented in this study.

**DNA samples preparation**. The The genomic DNAs were extracted using standard-well known protocols (i.e., phenol-chloroform extraction followed by isopropanol/ethanol purification) and the purity of the DNA was confirmed by measuring $A_{260/280}$ absorbance ratio. Briefly, the cells were suspended in lysis buffer to lyse and release the nucleic acids and proteins into the solution. To remove the protein and RNA in the solution a digestion step was performed using proteinase and RNase enzymes, respectively. The digested proteins and RNA were removed by phenol chloroform solvent extraction and the DNA was purified by isopropanol/ethanol precipitation. Short DNA fragments were prepared by amplifying (Forward primer: 5′-ATTCAGTCCACAACAAYGTTGGTTGAG TTTATAAGTAGGGATAGT-3′ and Reverse Primer: 5′-ACRACCRCAACAA CCAAACCCT-3′) a bisulphite treated 140 bp region of EN1 gene using deoxycytidine triphosphate (dCTP) for unmethylated and methyl dCTP for methylayted DNA. Whole Genomic Amplified (WGA) DNA samples were prepared by amplifying 50 ng of BT474 breast cancer cell derived DNA using a REPLI-g whole genome amplification kit (QIAGEN Pty. Ltd., Venlo, Netherlands) as per manufacturer's instruction. Jurkat 100% methylated and Azacytidine treated Jurkat gDNA were purchased from New Englands Biolabs. Enzymatically methylated gDNAs are obtained using the M.SssI CpG methyltransferase enzyme (New England Biolabs), which was allowed to insert methyl groups onto CpG sites of unmethylated whole genome amplified (WGA) DNA in the presence of SAM donor, according to manufacturer instructions. Reaction was performed for periods of 2 min, 5 min, 10 min, 15 min, 30 min or 60 min and then stopped by thermal inactivation of the enzyme at 65 degrees. A negative control is obtained by performed reaction with previously inactivated enzyme (0 min reaction). DNA methylation levels are confirmed by restriction enzymatic digestion using methylation sensitive HpaII restriction enzyme (New England Biolabs). DNAs from tissues were extracted by following standard procedure. Briefly, formalin fixed paraffin embedded (FFPE) or PAXgene fixed paraffin embedded (PFPE) tissues were first treated with xylene to remove the paraffin. After washing with ethanol, the tissues were vacuum dried followed by complete overnight digestion with Proteinase K. The DNAs were then extracted by either using Qiagen kit or following standard Phenol-Chloroform extraction procedure. The cfDNAs were extracted from plasma by using standard protocol. Briefly, 1 mL plasma was mixed with 100 μL of 250 mM EDTA and 750 mM NaCl solution followed by the addition of 100 μL of 100 g/L sodium dodecyl sulphate. To digest the protein in plasma, 20 μL of Proteinase K was then added to the mixture. The plasma solution was incubated at 56 °C for 2 h and 6 M NaCl was used to precipitate the protein. Finally, the supernatant was taken for phenyl chloroform extraction and isopropanol precipitation of cfDNA. Since the clinical tumour samples used in the above experiment were extracted from paraffin embedded formalin fixed (FFPE) tissues or PAXgene tissue blocks, which is a process that may cause a certain degree of DNA degradation that may not be present in gDNA from healthy individuals, we investigated the size fragmentation profile of a subset of cancerous epigenomes using the Agilent 2100 Bioanalyzer (High Sensitivity DNA chip). All the DNAs were found quite integrate and scarcely fragmented (See Supplementary Figure 8b). To further test the effect of DNA degradation on their adsorption behaviour, we

sonicated our genomic BT474 DNA for one minute and measured the adsorption level of the degraded BT474 DNA. As shown in supplementary figure 8a, the degradation has little effect on the DNA adsorption towards gold surface.

**Global DNA methylation analysis**. Global methylation analysis of BT474 DNA was performed by using Imprint® Methylated DNA Quantification kit from Sigma Aldrich as per manufacturer instructions. Briefly, the desired amount of DNA was diluted in 30 μL DNA Binding Solution and added to each well of the plate. The DNA Binding Solution alone was used as a blank. The wells were covered and the samples were incubated at 37 °C for 60 minutes. After incubation, 150 μL of Block Solution was directly added to each well and incubated again for 30 minutes. All the solution from each well was then removed and the wells were washed three times with 150 μL of 1x Wash Buffer. Methylation specific capture antibody was then diluted in 50 μL wash buffer, added to each well and incubated for 60 minutes. After that the capture antibody solution was removed from each well and the wells were washed four times with 150 μL wash buffer. Subsequently, the diluted Detection Antibody was added to each well and incubated, removed and washed. After that 100 μL of Developing Solution was added to each well and the wells were incubated at room temperature away from light for 1–10 minutes. When the solution turned blue, 50 μL of stop solution was added to each well and the solutions were turned yellow. The absorbance of the solutions in each well was then measured at 450 nm by using a plate reader. The global methylation level of all DNAs is calculated using following equation.

$$\text{Global Methylation level} = \left[ \left( A_{450}\text{Sample} - A_{450}\text{Blank} \right) / \left( A_{450}\text{Methylated Control DNA} - A_{450}\text{Blank} \right) \right] \times 100 \quad (1)$$

**Electrochemical detection**. All electrochemical experiments were carried out using a CH1040C potentiostat (CH Instruments) with a three-electrode system consisting of a gold working electrode (2 mm in diameter), Pt counter electrode, and Ag/AgCl reference electrode (all electrodes are from CH Instruments, USA). Differential pulse voltammetric (DPV) experiments were conducted in 10 mM PBS solution containing 2.5 mM [K$_3$Fe(CN)$_6$] and 2.5 mM [K$_4$Fe(CN)$_6$] electrolyte solution. DPV signals were obtained with a potential step of 5 mV, pulse amplitude of 50 mV, pulse width of 50 ms, and pulse period of 100 ms. For DNA methylation detection, the gold electrodes were initially cleaned by polishing with Alumina polishing powder (CH Instruments) followed by ultra-sonication with acetone and deionised water for 5 min and then dried under the flow of nitrogen. DPV signals of clean electrodes were measured in electrolyte solution to get the baseline current. Purified gDNA (5 μL of 10 ng/μL concentration in SSC 5X buffer at neutral pH) was then adsorbed onto gold electrodes for 10 min. Subsequently, the adsorption competence was measured by Differential Pulse voltammetry (DPV) in presence of the [Fe(CN)$_6$]$^{3-/4-}$ redox system. Upon DNA adsorption, the coulombic repulsion between negatively-charged ferrocyanide ions in the buffer and negatively-charged DNA phosphate groups on the electrode surface partially hinder the diffusion of ferrocyanide ions to the electrode surface. This generates a Faradaic current signal, which is proportionally lower than the bare electrode signals as increasing numbers of DNA molecules become adsorbed onto the surface[25–27] (i.e., the greater the DNA adsorption is, the larger the relative current signal difference, %$i_r$, with respect to the original baseline. The relative adsorption currents (i.e., %$i_r$, % difference of the DPV signal generated for DNA sample ($i_{\text{sample}}$) with respect to the baseline current ($i_{\text{baseline}}$)) due to the adsorption of DNA samples were then measured by using equation 2.

$$\text{Adsorption current } (\%i_r) = \left[ \left( i_{\text{baseline}} - i_{\text{sample}} \right) / i_{\text{baseline}} \right] \times 100 \quad (2)$$

**Detection by AuNP system**. Experiments are performed using 8.5 μL of 10 nm Tannic-capped AuNPs (Sigma), which were mixed with 1 μL of DNA samples (i.e. genomic DNA at 50 ng/μL concentration or cfDNA at 1 pg/μL concentration). Aggregation was achieved by addition of 1.5 μL of SSC 5X. Absorbance ratio at 520/658 was measured using Nanodrop to quantify the shift of surface plasmon band due to the adsorption of DNA and aggregation of AuNPs.

**TEM measurements**. Experiments were performed using a Jeol 1010 or Hitachi HT 7700 transmission electron microscope (TEM) at 100 kV. Equal amounts of normal and cancerous DNA were spotted and dried onto 400 mesh square carbon grids coated with formvar (Proscitech) and then stained with ammonium molybdate (1%, pH 7) for imaging purposes.

**Atomic force microscopy**. DNA Samples (10 ng/μL in SSC 5x buffer) were adsorbed on ultraflat gold surface for 20 min and then AFM Experiments are performed with Cypher AFM system (Asylum Research) on air tapping-mode with a 30 nm radius sharp silicon tip.

**Reporting Summary**. Further information on research design is available in the Nature Research Reporting Summary linked to this article.

 

## Data availability

The data that support the findings are available on request. A reporting summary for this article is available as a Supplementary Information file.

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

## Acknowledgements

This work was supported by the UQ Postdoctoral Research Fellowship (2012001456) awarded for LGC, ARCDP (DP120102503, DP180102868) and National Breast Cancer Foundation of Australia (CG-08-07 and CG-12-07) for M.T. These grants have significantly contributed to stimulate the environment of the research described here. We also thank Prof. Peter Schmid and Dr. Alice Shia for the generous donation of samples from UK-Forever clinical trial (plasma and tissue samples from ER+ breast cancer patients). We also acknowledge the support from Master's students Ting-Yun Lin, Adithi Dinesh Nambiar and Yefei Liang in sample preparation. M.T., A.A.I.S., L.G.C. acknowledge Alexandra Cooksley Rowe (1976–2014) for inspiration.

## Authors contributions

L.G.C., A.A.I.S. and M.T. conceived the project. L.G.C. and A.A.I.S. designed the experiments with the assistance of M.T. D.K. and A.A.I.S. extracted DNAs from patient samples. M.J.A.S. helped in designing electrochemistry experiments. A.A.I.S., L.G.C., Z.L. and A.W. performed most of the experiments. Y.S.G. performed TEM and AFM images. R.A.G., H.S., M.K.G. and R.J.S. provided the samples from prostate, lymphoma and colorectal cancer respectively and their associated clinical information. L.G.C. and A.A.I.S. wrote the paper with the contribution from all authors.

## Additional information

**Competing interests:** The authors declare no competing interests.

