## [Peer Review File · Nature Communications]

Reviewers' comments:

Reviewer #1 (Remarks to the Author):

This manuscript reports on exploiting DNA physicochemical properties to develop cancer detection assays that are potentially simple and powerful. There are three aspects to the findings – the physicochemical properties/experiments (which I cannot comment on), the link to DNA methylation, and the clinical detection link. Overall, the test is provocative in its simplicity and detection power. There are some major issues however that need to be clarified and might lower enthusiasm for the paper depending on findings.

1. Throughout, the manuscript does not address DNA quality or average fragment size as a variable in the assays. Surely, the physicochemical properties of PCR products might be different from large native DNA fragments and, in turn, degraded DNA (as is often seen in cell free preparations and DNA extracted from primary samples) might behave differently from intact DNA? This is a critical variable that needs to be carefully addressed.

2. Throughout, the manuscript suffers from a lack of clarity about the methylation-assay results link. Indeed, the “methyscape” of normal vs cancer is not as described. Normal tissues are the ones that have large tracts of uniformly methylated and unmethylated DNA. Cancer cells have large tracts of variably demethylated DNA (with a high degree of heterogeneity) and variably hypermethylated CpG islands (uniformity in CpG island hypermethylation is only seen in cell lines, not primary tumors). Setting aside discussions of “methyscapes”, the interpretation of a methylation/assay link is based on a small number of samples and often lacks controls (in Fig. 2b, why study HCT116DKO but not HCT116? Why study JURKATT AZA but not JURKATT control?). To be convinced of the link, one needs to see many more samples studied with careful controls, dilutions and admixtures.

3. The clinical data (Fig. 3) are quite interesting. Most promising is Figure 3d but it is based on only 20 samples and 20 controls. I would suggest validation of this assay in a substantially larger data set, making sure that the samples are randomized and blinded to the operator, and that the cases and controls are matched for age, gender, DNA concentration and DNA quality (if possible). Age and gender matching is essential given changes in DNA methylation with age and on the inactive X. I would also suggest that the authors estimate positive and negative predictive values (taking into account prevalence and, if possible, stage of the disease). This validation should also be done for Fig. 4e.

Reviewer #2 (Remarks to the Author):

Title: Epigenetically reprogrammed methylation landscape drives the self-assembly of DNA and can serve as a universal cancer biomarker

Authors: Abu Ali Ibn Sina , Laura G. Carrascosa , Muhammad J. A. Shiddiky, Ziyu Liang, Andri Wardiana, Yadveer S. Grewal, Darren Korbie, and Matt Trau

Overview: This contribution describes the characterization and detection of methylated DNA with the ultimate goal of diagnosing cancer in a non-invasive way. The contribution initially describes the characterization of genomic DNA adsorbing onto surfaces using TEM and AFM. The TEM study was performed to show that genomic DNA clusters together on a surface differently in its wild-type non-cancerous state compared to when methylation patterns change. Non-methylated or regular wild type genomic DNA formed smaller clusters on the surface compared to differently methylated – i.e cancerous or special control DNA. AFM was shown to study how this interaction occurs on a gold surface, with the finding that extensively (i.e. 100%) cytosine methylation caused the DNA to not adsorb as strongly – in a similar fashion to regular genomic DNA. However, DNA from cancerous lines adsorbed much more to the surface.

The authors used these studies to explore two detection strategies – one electrochemical and one optical – in order to test the adsorption performance on gold electrodes and the ability of the DNA to cause AuNPs to aggregate, respectively. DNA from cancerous cells that exhibited differential methylation patterns, termed the “methyloscape,” generated higher adsorption to gold electrodes, which was measured in the decrease in electrochemical current from a redox active solution species (ferricyanide couple). Higher adsorption of negatively charged DNA caused the repulsion of the redox couple and lower signals. Similar effects were seen using AuNPs to detect methylated DNA via the clustering of the nanoparticles and the shift in absorbance maximum. The authors were able to demonstrate that genomic DNA extracted from cancerous cells or from plasma from cancer patients could be detected as compared to that originating from non-cancerous samples in a statistically significant manner.

Comments: This contribution is powerful. The applications are simple, yet the experiments are thoughtful and thorough. Almost every perceivable question was addressed through the document. The authors show that a simple strategy of adsorbing DNA to an electrode can be used to provide a quick means to assay for cancer. The authors provide elegant controls to show that it's not the sheer amount of DNA methylation that matters, rather, the islands of CpG methylation that change as a cell transitions from benign to malignant is important in the adsorption of the DNA to the surfaces. This ultimately impacts detection. The authors do an admirable job discussing the mechanisms behind this process. Overall, this is a very high impact contribution needing almost nothing in the way of revisions to be published. Some proofreading might be necessary. For instance some sentences have odd subject/verb usage “Interestingly, agglomerated fully methylated Jurkat DNA (Image IV) was appeared to be able to interact with AuNPs” However, these are very minor issues. This is an excellent contribution that will surely find a wide-ranging audience and should be published.

Reviewer #3 (Remarks to the Author):

Sina et al. analyze consequences of DNA methylation on DNA physicochemical properties. The major hypothesis is that different methylation landscapes of normal and cancer epigenomes impact the respective physicochemical and self-assembly properties in aqueous solutions, which can be measured by their interactions with solid surfaces. They find that key physicochemical properties of normal genomic DNA differs from epigenetically reprogrammed cancer genomes and that hydrophobic properties of methylcytosines, depending on both their levels and patterning, resulting in different DNA polymer conformations in solution are a major cause. Furthermore, the different solvation properties of cancer and normal epigenomes affect their affinity to metal surfaces, such as gold. Hence, a focus of the study was to characterize the affinity of genomic DNA towards gold in relation to the methylation level and patterning. The authors combined electrochemical and colorimetric techniques to detect methylation biomarkers from cancer genomes based on adsorption on planar and colloidal gold surfaces and tested how these features can reflect clinicopathological states.

This is a very interesting paper, which may pave the way for novel strategies to analyze epigenetic changes. The authors demonstrate convincingly that methylation can impact the self-assembly of genomic DNA in solution -i.e., the larger the number of methylcytosines in the genome, the greater are the chances for DNA to form aggregates in aqueous solution.

However, enthusiasm about the paper is diminished by lack of controls so that it is not possible to judge whether the conclusions are justified. Hence, there remain a number of issues to be addressed:

1. The authors claim that the BT474 breast cancer cell line has “approximately 43% global methylation levels” and refer to the method section for details. In the method section, the kit used for methylation analysis (Imprint Methylated DNA Quantification kit from Sigma Aldrich) is described; however, how the number of 43% was derived is not shown.
2. On page 6 (the manuscript does not have page numbers...) it is claimed “Additional TEM experiments and technical replicates of different biological samples were also performed”; however, where are these additional experiments? For sure not in Fig. S3 to which the authors refer.
3. The electrochemical detection is hard to understand from the main text alone. It would be helpful if general statements, such as “...the greater the DNA adsorption is, the larger the relative current signal difference, %ir,...” would be moved from the method section to the main text.
4. Figure 2a is not helpful; the legend needs to be extended in order to understand this figure.
5. Similar to the first point: “...we also tested the adsorption competence of the epigenome of HMEC-CA cell line. This cell line was a version of normal HMEC cell lines that had undergone a limited number of cell-passages. With each successive cell passage, the epigenome suffered gradual reprogramming leading to spontaneous partial transformation into a premalignant cell type. Notably, gDNA derived from HMEC-CA cell line showed higher surface adsorption than the normal HMEC control cells (Fig. 3a).”: this is not convincing; this cell line is not common and a detailed analysis of the methylation status is missing.
6. On page 11 the authors write: “we analyzed 72 epigenomes extracted from patient tumor tissues”; however, where is this characterization? The manuscript shows gDNA adsorption levels but the methylation levels were not confirmed by another technology, e.g. whole-genome bisulfite sequencing.
7. For all cancer samples there are no clinical information provided. However, at least basic data on tumor stage, metastasis status and so on, maybe in form of a table, are needed.
8. In the main text, the authors write that cfDNA was isolated from plasma samples. However, in the method section they write “cfDNA were extracted from serum...”. There are fundamental differences between plasma and serum and it has to be clarified what was actually used.
9. The cfDNA part is highly interesting, but again basic clinical information is missing from the breast cancer patients (e.g. localized vs. metastatic disease). No control experiments with other means were conducted to establish the actual ctDNA variant allele frequency (VAF). In most plasma samples from cancer patients, the ctDNA VAF is below 10%, in a significant proportion even below 1%. Hence, the actual ctDNA VAF has to be determined with established methods (e.g.

digital PCR) in order to judge the real performance of the approach described in this manuscript. In fact, it is questionable whether the methylation based approach will work with an “average plasma sample” (i.e. one with low ctDNA VAF) from a patient with cancer.

10. Due to the high fragmentation of plasma DNA it is unlikely that cluster of methylated and unmethylated regions contribute to the different current values. The authors themselves discuss that ultrashort cfDNA fragments may cause the differences instead. The real reason should be elucidated.

11. Within the cfDNA part the authors make the very bold statement: “...our approaches outperformed most of the current techniques for cancer detection including sequencing...” Unfortunately, because of the aforementioned reasons the presented data do not allow concluding whether this is true.

12. The “naked eye detection of cancer” part is spectacular; however, it is just stated that 21 epigenomes from different cancer types were evaluated. Without any clinical information and more detailed description of these epigenomes the performance cannot be judged.

13. Are the 20 cfDNA samples in Fig. 3d identical to those from Fig. 4e? Similar to the comments above: further information on disease status and ctDNA VAF are needed!

14. Figure 4a: “Healthy DNA” should be avoided. It is clear what the authors mean, but there is no “healthy DNA”.

15. Page 19: “... our approaches showed large potential for cancer diagnosis as evidenced by the ROC graphs...”. This may be true, but as mentioned before, it is hard to judge without clinical information. Furthermore, the information cancer or not can be easily obtained –and more accurately- by other established routine diagnostic means. An evaluation of resolution limits is completely missing.

In summary, this is a fascinating paper suggesting that physicochemical properties between cancer and normal DNA may enable detection of methylscape biomarker in a single step based on interfacial biosensing strategy. In theory, this approach may have a great potential and may facilitate real time clinical decision-making but much more detailed work with an establishment of resolution limits is needed.

Detailed Responses to Reviewers' comments

Reviewer #1

This manuscript reports on exploiting DNA physicochemical properties to develop cancer detection assays that are potentially simple and powerful. There are three aspects to the findings – the physicochemical properties/experiments (which I cannot comment on), the link to DNA methylation, and the clinical detection link. Overall, the test is provocative in its simplicity and detection power. There are some major issues however that need to be clarified and might lower enthusiasm for the paper depending on findings.

Reply: We are delighted with reviewer's view of our test as "provocative in its simplicity and detection power", and apologize for the lack of clarity in the manuscript, which may have contributed to some of his/her concerns. We have now addressed the reviewer's comments below and modified the manuscript accordingly.

Specific Comments

- 1. Throughout, the manuscript does not address DNA quality or average fragment size as a variable in the assays. Surely, the physicochemical properties of PCR products might be different from large native DNA fragments and, in turn, degraded DNA (as is often seen in cell free preparations and DNA extracted from primary samples) might behave differently from intact DNA? This is a critical variable that needs to be carefully addressed.*

Reply: We agree with the reviewer on the important role of DNA quality for Methylation analysis. Regarding DNA quality, we have mentioned in the DNA samples preparation part of the Methods Section (Page 23, red marked) that the DNA has been thoroughly purified and the quality has been checked by measuring $A_{260/280}$ absorbance ratio. Regarding average fragment size, there is a chance that samples that have been stored for longer times or subjected to paraffin procedures may be more degraded and have a different response than control samples not subjected to this process. For this reason, we mentioned in the DNA samples preparation part of the Methods Section (Page 24, red marked) that we have tested the integrity of our DNA genomes, and found that both normal and cancer derived ones were all of high molecular weight (fragment sizes are $\approx 10\text{kb}$) and not significantly fragmented. Hence, the integrity of the DNA could not be a reason for explaining the different response in cancer vs control adsorption trends. Additionally, we have now tested the fragment size of more DNA samples and found that all the samples were high molecular weight. The additional data has been added to the supplementary Fig. S8 of the revised manuscript.

- 2. (a) Throughout, the manuscript suffers from a lack of clarity about the methylation-assay results link. Indeed, the "methyscape" of normal vs cancer is not as described. Normal tissues are the ones that have large tracts of uniformly methylated and unmethylated DNA. Cancer cells have large tracts of variably demethylated DNA (with a high degree of*

heterogeneity) and variably hypermethylated CpG islands (uniformity in CpG island hypermethylation is only seen in cell lines, not primary tumors).

Reply: This is an insightful comment. We agree with the reviewer in the complexity of describing the “methyscape” of normal and cancer genomes and apologise if the description we made was too simplistic. Our aim was to highlight that genomes from adult normal tissues tend to exhibit overall higher degrees of methylation, which are also quite evenly dispersed (uniform) throughout the genome. In contrast, this distribution changes drastically during the course of cancer—as DNA gradually loses most of that methylation and commences to exhibit more defined methylated areas, where methylated sites are heavily clustered within a short span. However, we fully agree with the reviewer that, within this averaged trend, there is an intrinsic heterogeneity in the methylation patterns across cells within the tissue—particularly in the context of cancer.

We would like to note that despite this heterogeneity, the drastic changes in the cell’s “methyscape” during cancer progression is well documented in the literature as a key feature of cancer epigenetics (Smith, Z.D. & Meissner, A. *Nature Reviews Genetics* **14**, 204-220 (2013); Suzuki, M.M. & Bird, A. *Nature Reviews Genetics* **9**, 465-476 (2008)). It is this global change in the methylation pattern, and overall levels and distribution that our methodology is able to detect in a highly simplified way.

We also agree with the reviewer that the DNA from normal cells has large tracts of uniformly methylated regions only interrupted by blocks of unmethylated regions. This fact indeed supports our Methyscape hypothesis. We hypothesize that due to the large tracts of uniformly methylated regions in normal DNA, large number of hydrophobic methyl groups in solution come into proximity with each other and collapse into self-contained nano- and micro-sized domains surrounded by hydrophilic unmethylated regions, whose surface would then have the same properties and adsorption affinity as a fully unmethylated DNA (described in the “Proposed Mechanism” section, page 20 of the manuscript).

In the same line, the fact that cancer cells have large tracts of variably demethylated DNA (with a high degree of heterogeneity) and variably hypermethylated CpG islands are also in agreement with our hypothesis. Despite some degree of variable demethylation across the genome, the significant reduction in the overall methylation levels compared to normal genomes, would reduce overall hydrophobicity of the DNA colloid and the chances for DNA to collapse into the above-described self-contained nano- and micro-sized domains. This, in turn, would contribute to increase its overall solubility in aqueous solutions and the chances for hyper-methylated CpG islands to be more accessible and exposed for interacting with the gold surface.

To address the reviewer’s comment, we have now added the following section to the page 19 of the revised manuscript.

“The work presented here is based on the detection of a global methylation landscape in cancer which we referred to as Methyscape. The Methyscape in cancer involves a drastic change in the global methylation levels and patterning of methylation marks, when cancer and normal derived DNAs are compared at the whole-genome level. The

Methylation Landscape (i.e., Methyscape) of normal DNA comprises of large tracts of uniformly methylated DNA separated by blocks of unmethylated DNA. In contrast, Methyscape of cancer cells have large tracts of variably demethylated and variably hypermethylated CpG islands. Despite relatively high degree of heterogeneity in methylation levels and pattern in cancer-derived DNA, the drastic differences between the respective Methyscape of normal and cancer-derived DNAs is an evident hallmark of the cancer process. This difference is already well documented in the literature and, along with the data presented in this manuscript, provides the foundations for considering this phenomenon as a general biomarker for cancer.”

2 (b) *Setting aside discussions of “methylandsapes”, the interpretation of a methylation/assay link is based on a small number of samples and often lacks controls (in Fig. 2b, why study HCT116DKO but not HCT116? Why study JURKATT AZA but not JURKATT control?). To be convinced of the link, one needs to see many more samples studied with careful controls, dilutions and admixtures.*

Reply: With regard to this comment, we would like to note that the data for HCT116 cell derived DNA was already present in *Figure 3a* of the manuscript. The purpose of figure 2b was to show how the adsorption pattern of genomic DNA changes with their global methylation levels. However, we agree with the reviewer that some additional control data will better clarify the concept. Thus, we have included the data for Jurkat cancer DNA and performed additional experiments to quantify the global DNA methylation levels of Jurkat and HMEC DNA (See Fig. 2b of the modified manuscript). These data now clearly show the changes in the adsorption trend of genomic DNAs with their overall methylation levels. Regarding HCT116DKO cell derived DNA, unfortunately we consumed all the materials of this sample and therefore couldn't measure its global methylation levels. Hence, to avoid problems, we have removed this data from the manuscript. This data is not essential for our conclusions.

2. *The clinical data (Fig. 3) are quite interesting. Most promising is Figure 3d but it is based on only 20 samples and 20 controls. I would suggest validation of this assay in a substantially larger data set, making sure that the samples are randomized and blinded to the operator, and that the cases and controls are matched for age, gender, DNA concentration and DNA quality (if possible). Age and gender matching is essential given changes in DNA methylation with age and on the inactive X. I would also suggest that the authors estimate positive and negative predictive values (taking into account prevalence and, if possible, stage of the disease). This validation should also be done for Fig. 4e.*

Reply: As suggested by the reviewer, we have now performed additional experiments including a sample size of 100 cancer patients with age, gender and DNA concentration (10ng/μl for tissue derived gDNA and 1pg/μl for plasma cfDNA) matched controls and their identity was blinded to the operator (See main text Fig. 3d, 4e, and supplementary Fig. S12, S13, S24 and S25 for details). The clinical information about the samples such as gender, age,

cancer type and status are included in the supplementary Table, S10, S11 and S12 of the revised manuscript. We have also performed statistical diagnostic efficacy test and calculated sensitivity, specificity, accuracy and positive and negative predictive values for both electrochemistry and nanoparticle based techniques. These information are now included in the main text Fig. 3c, 3d, 4d, 4e and supplementary table S2, S3, S4 and S5 of the revised manuscript.

Reviewer #2

Overview: *This contribution describes the characterization and detection of methylated DNA with the ultimate goal of diagnosing cancer in a non-invasive way. The contribution initially describes the characterization of genomic DNA adsorbing onto surfaces using TEM and AFM. The TEM study was performed to show that genomic DNA clusters together on a surface differently in its wild-type non-cancerous state compared to when methylation patterns change. Non-methylated or regular wild type genomic DNA formed smaller clusters on the surface compared to differently methylated – i.e cancerous or special control DNA. AFM was shown to study how this interaction occurs on a gold surface, with the finding that extensively (i.e. 100%) cytosine methylation caused the DNA to not adsorb as strongly – in a similar fashion to regular genomic DNA. However, DNA from cancerous lines adsorbed much more to the surface.*

The authors used these studies to explore two detection strategies – one electrochemical and one optical – in order to test the adsorption performance on gold electrodes and the ability of the DNA to cause AuNPs to aggregate, respectively. DNA from cancerous cells that exhibited differential methylation patterns, termed the “methyscape,” generated higher adsorption to gold electrodes, which was measured in the decrease in electrochemical current from a redox active solution species (ferricyanide couple). Higher adsorption of negatively charged DNA caused the repulsion of the redox couple and lower signals. Similar effects were seen using AuNPs to detect methylated DNA via the clustering of the nanoparticles and the shift in absorbance maximum. The authors were able to demonstrate that genomic DNA extracted from cancerous cells or from plasma from cancer patients could be detected as compared to that originating from non-cancerous samples in a statistically significant manner.

Comments: *This contribution is powerful. The applications are simple, yet the experiments are thoughtful and thorough. Almost every perceivable question was addressed through the document. The authors show that a simple strategy of adsorbing DNA to an electrode can be used to provide a quick means to assay for cancer. The authors provide elegant controls to show that it's not the sheer amount of DNA methylation that matters, rather, the islands of CpG methylation that change as a cell transitions from benign to malignant is important in the adsorption of the DNA to the surfaces. This ultimately impacts detection. The authors do an admirable job discussing the mechanisms behind this process. Overall, this is a very high impact contribution needing almost nothing in the way of revisions to be published. Some proofreading might be necessary. For instance some sentences have odd subject/verb usage “Interestingly, agglomerated fully methylated Jurkat DNA (Image IV) was appeared to be able to interact with AuNPs” However, these are very minor issues. This is an excellent*

contribution that will surely find a wide-ranging audience and should be published.

Reply: We are delighted that Reviewer 2 recognised the innovation, high significance, technical quality and broad interest of the work presented in our manuscript. We also thank Reviewer 2 for considering the work as a powerful contribution and an admirable job. As suggested by the reviewer, the manuscript has been thoroughly checked for any typographical error.

Reviewer #3

Sina et al. analyze consequences of DNA methylation on DNA physicochemical properties. The major hypothesis is that different methylation landscapes of normal and cancer epigenomes impact the respective physicochemical and self-assembly properties in aqueous solutions, which can be measured by their interactions with solid surfaces. They find that key physicochemical properties of normal genomic DNA differs from epigenetically reprogrammed cancer genomes and that hydrophobic properties of methylcytosines, depending on both their levels and patterning, resulting in different DNA polymer conformations in solution are a major cause. Furthermore, the different solvation properties of cancer and normal epigenomes affect their affinity to metal surfaces, such as gold. Hence, a focus of the study was to characterize the affinity of genomic DNA towards gold in relation to the methylation level and patterning. The authors combined electrochemical and colorimetric techniques to detect methylation biomarkers from cancer genomes based on adsorption on planar and colloidal gold surfaces and tested how these features can reflect clinicopathological states. This is a very interesting paper, which may pave the way for novel strategies to analyze epigenetic changes. The authors demonstrate convincingly that methylation can impact the self-assembly of genomic DNA in solution -i.e., the larger the number of methylcytosines in the genome, the greater are the chances for DNA to form aggregates in aqueous solution. However, enthusiasm about the paper is diminished by lack of controls so that it is not possible to judge whether the conclusions are justified. Hence, there remain a number of issues to be addressed:

Reply: We are pleased with the reviewer's view of our manuscript as "*a very interesting paper, which may pave the way for novel strategies to analyze epigenetic changes*", and would like to thank him/her for their insightful suggestions. We believe that we have now incorporated the required additional data and explanations to address reviewer's concerns and improved significantly the quality and scientific merit of our work.

Specific Comments

- 1. The authors claim that the BT474 breast cancer cell line has "approximately 43% global methylation levels" and refer to the method section for details. In the method section, the*

kit used for methylation analysis (Imprint Methylated DNA Quantification kit from Sigma Aldrich) is described; however, how the number of 43% was derived is not shown.

Reply: We apologise for the lack of clarity about the global methylation levels of BT474 DNA. This kit includes standards of known methylated levels to calibrate kit's response and accurately estimate the methylation levels of tested samples. We have now included a more detailed explanation in the method section (Page 24, red marked) showing how methylation levels from the BT474 cell line were determined.

2. *On page 6 (the manuscript does not have page numbers...) it is claimed "Additional TEM experiments and technical replicates of different biological samples were also performed"; however, where are these additional experiments? For sure not in Fig. S3 to which the authors refer.*

Reply: We apologise for not including the page numbers. This has now been rectified in the revised version. We also apologise for the lack of clarity in explaining the experiments depicted in Fig S3 and now we have included more images. Fig S3 has 12 TEM images which includes four different images of each WGA, BT474 and 100% Methylated Jurkat DNA. These are additional images, which were not used in the other figures of main text, correspond to different samples of the same DNA, and in most cases, were also analysed in different days. Hence, they do not represent different areas from within a unique sample. This information has been added to the revised manuscript (Page 6, red marked).

3. *The electrochemical detection is hard to understand from the main text alone. It would be helpful if general statements, such as "...the greater the DNA adsorption is, the larger the relative current signal difference, %ir,..." would be moved from the method section to the main text.*

Reply: As suggested by the reviewer, the electrochemical detection section is now clarified in the page 7 and 8 of the revised manuscript.

4. *Figure 2a is not helpful; the legend needs to be extended in order to understand this figure.*

Reply: As suggested by the reviewer, we have modified Fig 2 and the corresponding figure legend for better clarity.

5. *Similar to the first point: "...we also tested the adsorption competence of the epigenome of HMEC-CA cell line. This cell line was a version of normal HMEC cell lines that had undergone a limited number of cell-passages. With each successive cell passage, the epigenome suffered gradual reprogramming leading to spontaneous partial transformation into a premalignant cell type. Notably, gDNA derived from HMEC-CA cell line showed higher surface adsorption than the normal HMEC control cells (Fig.*

3a).”: *this is not convincing; this cell line is not common and a detailed analysis of the methylation status is missing.*

Reply: We agree with the reviewer that without genome-wide methylation data, the degree of methylation changes experienced in the *HMEC-CA* cell line with each passage cannot be confirmed. Unfortunately, we consumed all the materials of this sample and therefore couldn't measure its global methylation levels. Hence, to avoid problems, we have removed this data from the manuscript. This data is not essential for our conclusions.

6. *On page 11 the authors write: “we analyzed 72 epigenomes extracted from patient tumor tissues”; however, where is this characterization? The manuscript shows gDNA adsorption levels but the methylation levels were not confirmed by another technology, e.g. whole-genome bisulfite sequencing.*

Reply: We agree with the reviewer that the quantification of methylation levels of tested patient samples by another technique will further clarify the Methyscape mechanism. As suggested by the reviewer, we have now characterized 53 patient and 17 normal gDNA samples by analysing their global methylation levels using a whole genome methylation kit. Unfortunately, we had no leftover samples of 19 cancer patients and therefore couldn't characterize their methylation levels. These data are now included in the page 13/14 of the main text and supplementary Table S6 S7, S8 and S9 of the revised manuscript.

7. *For all cancer samples there are no clinical information provided. However, at least basic data on tumor stage, metastasis status and so on, maybe in form of a table, are needed.*

Reply: All the available clinical information of the samples is now included in the supplementary table S6, S7, S8, S9, S10, S11 and S12 of the revised manuscript.

8. *In the main text, the authors write that cfDNA was isolated from plasma samples. However, in the method section they write “cfDNA were extracted from serum...”. There are fundamental differences between plasma and serum and it has to be clarified what was actually used.*

Reply: We apologise for this error in describing the cfDNA source. All cfDNA were isolated from plasma samples. This is now corrected in the modified manuscript (Page 24).

9. *The cfDNA part is highly interesting, but again basic clinical information is missing from the breast cancer patients (e.g. localized vs. metastatic disease). No control experiments with other means were conducted to establish the actual ctDNA variant allele frequency (VAF). In most plasma samples from cancer patients, the ctDNA VAF is below 10%, in a significant proportion even below 1%. Hence, the actual ctDNA VAF has to be*

determined with established methods (e.g. digital PCR) in order to judge the real performance of the approach described in this manuscript. In fact, it is questionable whether the methylation based approach will work with an “average plasma sample” (i.e. one with low ctDNA VAF) from a patient with cancer.

Reply: We thank the reviewer for this excellent suggestion.

Regarding clinical information

We have now included the available clinical information of all the cfDNA samples in the supplementary table S10, S11 and S12 of the modified manuscript.

Regarding ctDNA variant allele frequency (VAF)

We agree with the reviewer that knowledge of the variant allele frequency (VAF) of the cfDNA samples would be valuable information to determine the sensitivity of our Methyscape application, and enable us to assess the total amount of tumour-derived material present in the cfDNA sample. Unfortunately, we consumed all the material to complete the Methyscape study and these samples are no longer available to determine the cfDNA variant allele frequency (VAF). However, to address the reviewer concern, we performed an additional experiment by spiking different proportion of (0.1%, 1%, 2.5%, 5%, 10%) a short and cluster methylated DNA template in a normal cfDNA solution. This data showed that our method can identify 1% cluster methylated template in presence of large amount of normal cfDNA sample, based on tests with control samples. This information is now included in page 15(red marked) and supplementary figure S18 of the revised manuscript.

10. Due to the high fragmentation of plasma DNA it is unlikely that cluster of methylated and unmethylated regions contribute to the different current values. The authors themselves discuss that ultrashort cfDNA fragments may cause the differences instead. The real reason should be elucidated.

Reply: We respectfully disagree with reviewer’s view. While cfDNA fragments are indeed short, with an average size of 165 bp, they vary between small fragments of 70 to 200 base pairs and large fragments of approximately 21 kilobases (Jahr, S. *et al. Cancer Res.* **61**, 1659–1665 (2001); Underhill, H.R. *et al. PLoS genetics* **12**, e1006162 (2016)). This size-range is sufficient for some of them to contain large motifs with a G + C content > 50% and a CpG frequency (observed/expected) > 0.6 hence, meeting the criteria of a CpG cluster, or CpG island. If methylated, these CpG rich fragments are expected to show high affinity to attach strongly to the gold surface. In contrast, fragments with no methylation, or with the pattern commonly seen in normal samples, where methylated sites are very dispersed—on average 1 methylated CpG every 150 bp—would essentially behave as non-methylated, and exhibit very low affinity towards the gold surface. We believe, this is the most plausibly explanation as to why, despite the short length, the different patterns between cancer, and

normal samples could still be elucidated by our technique. We have added this information to the page 15 of the modified manuscript to clarify this point.

11. Within the cfDNA part the authors make the very bold statement: "...our approaches outperformed most of the current techniques for cancer detection including sequencing..." Unfortunately, because of the aforementioned reasons the presented data do not allow concluding whether this is true.

Reply: As suggested by the reviewer we tried to provide more information to justify our hypothesis and rewrote the statement in the page 22 of the revised manuscript as below.

"We believe that this remarkably simple approach (i.e. Methyscape) with its excellent sensitivity and specificity would potentially be a better alternative to the current techniques for cancer detection including sequencing."

12. The "naked eye detection of cancer" part is spectacular; however, it is just stated that 21 epigenomes from different cancer types were evaluated. Without any clinical information and more detailed description of these epigenomes the performance cannot be judged.

Reply: We have now performed additional experiments with 100 Breast and colorectal cancer patient samples including age and gender matched study. These data are now included in main text Fig. 3d, 4e and supplementary Fig. S10, S11, S12, S13, S21, S22, S23, S24. Also the clinical information is included in the supplementary table S10, S11 and S12 of the modified manuscript.

13. Are the 20 cfDNA samples in Fig. 3d identical to those from Fig. 4e? Similar to the comments above: further information on disease status and ctDNA VAF are needed!

Reply: As per our previous reply above, the clinical information of cfDNA samples is now included in the supplementary table S10, S11 and S12 of the modified manuscript.

As replied above regarding ctDNA VAF study, we agree with the reviewer that knowledge of the variant allele frequency (VAF) of the cfDNA samples would be valuable information to determine the sensitivity of our Methyscape application, and enable us to assess the total amount of tumour-derived material present in the cfDNA sample. Unfortunately, we consumed all the material to complete the Methyscape study and these samples are no longer available to determine the cfDNA variant allele frequency (VAF). However, to address the reviewer concern, we performed an additional experiment by spiking different proportion of (0.1%, 1%, 2.5%, 5%, 10%) a short and cluster methylated DNA

template in a normal cfDNA solution. This data showed that our method can identify 1% cluster methylated template in presence of large amount of normal cfDNA sample, based on tests with control samples. This information is now included in page 15 (red marked) and supplementary figure S18 of the revised manuscript.

14. Figure 4a: “Healthy DNA” should be avoided. It is clear what the authors mean, but there is no “healthy DNA”.

Reply: As suggested, we have now replaced the term “Healthy DNA” with “Normal DNA”.

15. Page 19: “... our approaches showed large potential for cancer diagnosis as evidenced by the ROC graphs...”. This may be true, but as mentioned before, it is hard to judge without clinical information. Furthermore, the information cancer or not can be easily obtained –and more accurately- by other established routine diagnostic means. An evaluation of resolution limits is completely missing.

Reply: All the available clinical information is now included in the supplementary table S6, S7, S8, S9, S10, S11 and S12 of the revised manuscript. Although it is true that the presence of cancer can be obtained by other methods, we believe our approach has remarkable simplicity and sensitivity, which are the strengths in our method. For example, detection can be achieved in less than 10 min post-DNA extraction and requires very low amount of DNA input, particularly for cfDNA, which is not seen in any other available methodology. We have also performed statistical diagnostic test evaluation to calculate the sensitivity, specificity and positive and negative predictive values for both electrochemistry and naked eye techniques. These information are now included in the main text Fig. 3c, 3d, 4d, 4e and supplementary table S2, S3, S4 and S5 of the revised manuscript.

16. In summary, this is a fascinating paper suggesting that physicochemical properties between cancer and normal DNA may enable detection of methylscape biomarker in a single step based on interfacial biosensing strategy. In theory, this approach may have a great potential and may facilitate real time clinical decision-making but much more detailed work with an establishment of resolution limits is needed.

Reply: Thanks again for the recognition of the scientific merit and radical innovation of our work. We hope the additional experiments and changes we have made in the manuscript—as detailed in the above responses—now addressed all points raised.

Reviewers' comments:

Reviewer #1 (Remarks to the Author):

The paper is improved. I still have a couple of concerns:

1. Figure 1a is incorrect as I pointed out in my prior review. The schema for "cancer genome" is actually closer to what normal epigenomes look like. The response to reviews acknowledges this correctly (the difference between cancer and normal is in uniform normal vs. heterogeneous cancer) but the paper has not been modified much to reflect this – the whole cancer methylscape discussion is still based on the false premise. This does not affect the results – only the explanation for the results. While elegant, the proposed explanation is probably incorrect. I would point out that, were the authors correct, there should be large differences between male and female normal DNA due to X-inactivation. This hypothesis is actually testable using artificially methylated DNA ligated into unmethylated DNA). I would suggest toning down this discussion.
2. From the response to reviews, it appears the authors did not actually test their method on degraded vs non-degraded DNA of the same sample. The effects of degradation could easily be tested. This is a limitation that needs to be discussed.
3. The 1% cancer lower limit detection in cfDNA is also a limitation that needs to be discussed.
4. It could be worth mentioning in the discussion that while the AUCs for detection are promising, they are only promising in the context of a very easy test to do (e.g. analogous to fecal occult blood or CEA testing). The AUCs may not be promising enough to implement large screening programs and any positive result would have to be validated by another method.

Reviewer #3 (Remarks to the Author):

The authors have addressed all the comments and criticisms I have raised in my review of the original manuscript. However, there is still one major issue:

It is too bad that all the material was consumed to complete the Methylscape study and that the real cfDNA VAF cannot be established. Instead, the authors used short and cluster methylated template DNA and spiked this DNA in normal cfDNA solutions at different proportions. They illustrate the results in Figure S18 and conclude that "below 1% loading of methylated DNA" can be detected (page 15). However, in Figure S18 the relative adsorption for 1% (or less) methylated template DNA is well below 40%. According to Fig. 3d and the corresponding text a %ir => 35 was used as a kind of a threshold to distinguish between normal and cancer plasma samples. Hence the data suggest that low levels of tumor plasma DNA may not allow the diagnosis of cancer.

Furthermore, the addition of clinical data is appreciated. However, regarding the staging the authors may refer to <https://www.cancer.org/treatment/understanding-your-diagnosis/staging.html>. A grade as provided in Table S11 may affect the staging, but it cannot be equated with staging. This reviewer believes that it is likely that all patients had advanced diseases (at least the breast cancer patients had "metastatic disease"), hence the majority of patients had high ctDNA VAFs and that this explains the clear differences shown in Figure 3d. In any case, the resolution limits for the detection of cancer plasma DNA fragments were not determined in this study.

Minor point regarding the clinical data: colorectal cancer at the age of 23 years is unusual (sample number 1). If this patient had a heritable predisposition (FAP?) the authors may add a footnote so that readers know that this was not a typo.

There are some other issues, which may/should be addressed prior to publication:

1. Figure 4a still contains "healthy DNA".
2. Page 19: "The Methylation Landscape (i.e., Methylscape) of normal DNA comprises of large tracts of uniformly methylated DNA separated by blocks of unmethylated DNA. In contrast, Methylscape of cancer cells have large tracts of variably demethylated and variably hypermethylated CpG islands." Should it not be the other way round?
3. The increased sensitivity of short template DNA as compared to tissue derived genomic DNA is

intriguing. The authors may consider moving the discussion of this aspect (page 15, from the sentence starting with "Although the reason for..." till the end of the paragraph) to the section "Proposed mechanism of detecting methylscape biomarker".

Minor points: The manuscript has several typos (cfDNA is at some locations written as "cf-DNA", it should be "cfDNA" throughout the manuscript; legend Fig. 4d: "limphonode"; page 15: "CPG" instead of "CpG", and so on).

Detailed Responses to Reviewers' comments

Reviewer #1

The paper is improved. I still have a couple of concerns:

1. *Figure 1a is incorrect as I pointed out in my prior review. The schema for “cancer genome” is actually closer to what normal epigenomes look like. The response to reviews acknowledges this correctly (the difference between cancer and normal is in uniform normal vs. heterogeneous cancer) but the paper has not been modified much to reflect this – the whole cancer methylscape discussion is still based on the false premise. This does not affect the results – only the explanation for the results. While elegant, the proposed explanation is probably incorrect. I would point out that, were the authors correct, there should be large differences between male and female normal DNA due to X-inactivation. This hypothesis is actually testable using artificially methylated DNA ligated into unmethylated DNA). I would suggest toning down this discussion.*

Reply: We thank the reviewer for this suggestion and apologise for the lack of clarity regarding explaining the Methylscape of cancer vs normal genome. To further clarify the issue, we have modified Figure 1a and included the following section (adopted from our previous response to reviews as the reviewer liked it) on page 19 of the modified manuscript. Also to tone down the discussion, we have deleted the words like drastic, heavily and highly from the manuscript in explaining the Methylscape mechanism.

“Genomes from adult normal tissues tend to exhibit overall higher degrees of methylation, which are also quite evenly dispersed (uniform) throughout the genome. In contrast, this distribution changes during the course of cancer as DNA gradually loses methylation across the genome and exhibits more defined methylated areas where methylated sites are clustered within a short span. However, within this averaged trend, there is intrinsic heterogeneity in the DNA methylation patterns across cells within the tissue particularly in the context of cancer. Despite this heterogeneity, the changes in the cell’s DNA Methylation pattern and level during cancer progression are well documented in the literature as a key feature of cancer epigenetics. It is this global change in the methylation pattern, and overall levels and distribution that our methodology is able to detect in a simplified way and the data presented in this manuscript provides the foundations for considering this phenomenon as a general biomarker for cancer.”

We have also added the following statements in the proposed mechanism section for further clarity.

“Given this, we hypothesize that due to the large tracts of uniformly methylated regions in normal DNA, large number of hydrophobic methyl groups in solution come into proximity with each other and collapse into self-contained nano- and micro-sized

domains surrounded by hydrophilic unmethylated regions, whose surface would then have the same properties and adsorption affinity as a fully unmethylated DNA. The empirical data presented in this study supports this theory, and explains why a 100% methylated and heavily methylated normal epigenome have similar surface adsorption properties as a completely unmethylated sample (Fig. 2b and 3a). In the same line, the fact that cancer cells have large tracts of variably demethylated DNA (with a high degree of heterogeneity) with hypermethylated CpG islands are also in agreement with our hypothesis. Despite some degree of variable demethylation across the genome, the reduction in the overall methylation levels compared to normal genomes, would reduce overall hydrophobicity of the DNA colloid and the chances for DNA to collapse into the above-described self-contained nano- and micro-sized domains. This, in turn, would contribute to increase its overall solubility in aqueous solutions and the chances for hyper-methylated CpG islands to be more accessible and exposed for interacting with the gold surface.”

Regarding the X-inactivation

This is an insightful comment. We agree with the reviewer that due to the X chromosome inactivation, female normal DNA would be more methylated than the male DNA and thus there would possibly be a difference in adsorption level between male and female DNA. Our data in Figure 3B in fact support this hypothesis and showed that normal tissue derived gDNAs from female individual (for Breast tissue gDNA) have comparatively lower adsorption (average relative adsorption, $\%I_{r\text{Breast normal}} = 13.98$ vs $\%I_{r\text{Prostate normal}} = 20.82$) than the male individuals (prostate tissue gDNA). However, the methylation level for the normal tissue derived gDNAs from both female and male individuals are above the methylation maxima (i.e. the methylation level above which the DNA undergoes aggregation in solution). Thus we believe that above the methylation maxima, the difference in methylation level provides little effect in their adsorption towards gold surface and this difference is well below the adsorption level of patient DNA and thereby does not affect our results.

- 2. From the response to reviews, it appears the authors did not actually test their method on degraded vs non-degraded DNA of the same sample. The effects of degradation could easily be tested. This is a limitation that needs to be discussed.*

Reply: We have now performed an additional experiment as recommended by the reviewer which suggests that our results have little effect on DNA degradation. This is now included in the DNA samples preparation part of the Methods Section (Page 24 and 25, red marked) and the additional data has been added to the supplementary Fig. S8a of the revised manuscript.

- 3. The 1% cancer lower limit detection in cfDNA is also a limitation that needs to be discussed.*

Reply: To address this comment, we have included the following statement on page 15 (red marked) of the modified manuscript.

“This limit of detection may not be sufficient for very low levels of tumour cfDNA and with the current version of the method; we may not be able to detect cancer on a very early stage, as patients may have cancer DNA copies, usually expressed as mutant variant allele frequency (VAF), in levels below 1% in plasma”

- 4. It could be worth mentioning in the discussion that while the AUCs for detection are promising, they are only promising in the context of a very easy test to do (e.g. analogous to fecal occult blood or CEA testing). The AUCs may not be promising enough to implement large screening programs and any positive result would have to be validated by another method.*

Reply: As suggested by the reviewer we have included following statement on page 23 (red marked) of the modified manuscript.

“However, Methyscape in it’s current form is only able to determine the presence of disease and a detail analysis is required to fully understand the type, stage and disease recurrence.”

Reviewer #3

The authors have addressed all the comments and criticisms I have raised in my review of the original manuscript. However, there is still one major issue:

- 1. It is too bad that all the material was consumed to complete the Methyscape study and that the real cfDNA VAF cannot be established. Instead, the authors used short and cluster methylated template DNA and spiked this DNA in normal cfDNA solutions at different proportions. They illustrate the results in Figure S18 and conclude that “below 1% loading of methylated DNA” can be detected (page 15). However, in Figure S18 the relative adsorption for 1% (or less) methylated template DNA is well below 40%. According to Fig. 3d and the corresponding text a %ir => 35 was used as a kind of a threshold to distinguish between normal and cancer plasma samples. Hence the data suggest that low levels of tumor plasma DNA may not allow the diagnosis of cancer.*

Reply: We agree with the reviewer that the relative adsorption for 1% methylated template DNA is below 40%. Thus to address this limitation, we have added the following statement in the revised manuscript (page 15, red marked)

“As shown in the supplementary figure, S18, the relative adsorption of cfDNA increased with the increase of methylated template DNA in the solution and can detect low loading of methylated DNA fragments, but it was not possible to clearly detect levels in the range of 1% or less. This limit of detection may not be sufficient for very low levels of tumour cfDNA and with the current version of the method; we may not be able to detect cancer on a very early stage, as patients may have cancer DNA copies, usually expressed as mutant variant allele frequency (VAF), in levels below 1% in plasma”

2. *Furthermore, the addition of clinical data is appreciated. However, regarding the staging the authors may refer to <https://www.cancer.org/treatment/understanding-your-diagnosis/staging.html>. A grade as provided in Table S11 may affect the staging, but it cannot be equated with staging. This reviewer believes that it is likely that all patients had advanced diseases (at least the breast cancer patients had “metastatic disease”), hence the majority of patients had high ctDNA VAFs and that this explains the clear differences shown in Figure 3d. In any case, the resolution limits for the detection of cancer plasma DNA fragments were not determined in this study.*

Reply: We apologise for the confusion about the grade and stage. This is now rectified in the modified manuscript. In our highly focused proof of concept study, we have demonstrated the sensitivity of detecting cluster methylated DNA fragments in the range of 1-10%. We believe that once published, other technology enhancement will improve the sensitivity in the future.

3. *Minor point regarding the clinical data: colorectal cancer at the age of 23 years is unusual (sample number 1). If this patient had a heritable predisposition (FAP?) the authors may add a footnote so that readers know that this was not a typo.*

Reply: Regarding the age of this patient, we checked our data file and found that this is not a typo. However, as suggested by the reviewer, we have now added a footnote to clarify this issue for the readers (Supplementary Information page 30).

4. *Figure 4a still contains “healthy DNA”.*

Reply: This is now corrected in the revised manuscript.

5. *Page 19: “The Methylation Landscape (i.e., Methyscape) of normal DNA comprises of large tracts of uniformly methylated DNA separated by blocks of unmethylated DNA. In contrast, Methyscape of cancer cells have large tracts of variably demethylated and variably hypermethylated CpG islands.” Should it not be the other way round?*

Reply: We apologise for the lack of clarity regarding explaining the Methyscape of cancer vs normal genome. This is now clarified in the proposed mechanism section of the modified manuscript (page 19 and 20, red marked).

6. *The increased sensitivity of short template DNA as compared to tissue derived genomic DNA is intriguing. The authors may consider moving the discussion of this aspect (page 15, from the sentence starting with “Although the reason for...” till the end of the paragraph) to the section “Proposed mechanism of detecting methyscape biomarker”.*

Reply: We thank the reviewer for this suggestion. This section is now moved to the proposed mechanism section as suggested (page 21, red marked).

7. *Minor points: The manuscript has several typos (cfDNA is at some locations written as “cf-DNA”, it should be “cfDNA” throughout the manuscript; legend Fig. 4d: “limphonode”; page 15: “CPG” instead of “CpG”, and so on).*

Reply: we thank the reviewer for his thorough check-up and apologise for these typographical errors. The manuscript has now been checked thoroughly and the typos are corrected accordingly.

REVIEWERS' COMMENTS:

Reviewer #1 (Remarks to the Author):

The authors have satisfactorily addressed my comments.

Reviewer #3 (Remarks to the Author):

The authors have successfully addressed all outstanding issues.